# Identification of acquired Notch3 dependency in metastatic Head and Neck Cancer

Maria Kondratyev [1 ✉], Aleksandra Pesic[1], Troy Ketela[1], Natalie Stickle[2], Christine Beswick[1], Zvi Shalev[1], Stefano Marastoni [1], Soroush Samadian[1], Anna Dvorkin-Gheva[1], Azin Sayad[1], Mikhail Bashkurov[3], Pedro Boasquevisque[1], Alessandro Datti[3], Trevor J. Pugh [1], Carl Virtanen[2], Jason Moffat[4], Reidar A. Grénman[5], Marianne Koritzinsky[1] & Bradly G. Wouters [1 ✉]

During cancer development, tumor cells acquire changes that enable them to invade surrounding tissues and seed metastasis at distant sites. These changes contribute to the aggressiveness of metastatic cancer and interfere with success of therapy. Our comprehensive analysis of "matched" pairs of HNSCC lines derived from primary tumors and corresponding metastatic sites identified several components of Notch3 signaling that are differentially expressed and/or altered in metastatic lines and confer a dependency on this pathway. These components were also shown to be differentially expressed between early and late stages of tumors in a TMA constructed from over 200 HNSCC patients. Finally, we show that suppression of Notch3 improves survival in mice in both subcutaneous and orthotopic models of metastatic HNSCC. Novel treatments targeting components of this pathway may prove effective in targeting metastatic HNSCC cells alone or in combination with conventional therapies.

[1] Princess Margaret Cancer Centre University Health Network, Toronto, ON, Canada. [2] Princess Margaret Cancer Center, Bioinformatics and HPC Core, Toronto, ON, Canada. [3] SMART High-Content Screening facility at Network Biology Collaborative Centre, Toronto, ON, Canada. [4] Department of Molecular Genetics, University of Toronto, Toronto, ON, Canada. [5] Turku University Hospital, Turku, Finland. ✉email: mashlo88@gmail.com; Brad.Wouters@uhnresearch.ca

Metastatic cells acquire new and unique properties that permit them to invade tissues and seed metastases at distant sites[1,2]. In HNSCC, gene expression studies lead to identification of 102 genes that predict presence of lymph node metastasis in patients[3,4]. The first attempt to uncover genetic alterations that characterize metastatic HNSCC was performed by targeted sequencing of a collection of 53 recurrent and metastatic HNSCC tumors comparing to matched normal[5]. This study revealed several molecular alterations that were not previously reported in the primary tumors. Another study performed whole exome sequencing of 13 synchronous lymph node metastases and 10 metachronous recurrent tumors; notably, they were able to compare the advanced tumor samples to both matched normal controls and matched primary tumors from the same patients[6]. While 60–80% of mutations were shared between primary tumors and either lymph node metastasis or recurrences, several unique alterations were identified. Another group utilized whole exome sequencing as well as RNA sequencing to characterize a collection of HNSCC cell lines developed from primary tumors and matched metastases[7]. The authors confirmed that UM-SCC lines recapitulate most of the known genomic alterations reported for HNSCC; moreover, they discovered several novel mutations only present in metastatic/recurrent tumor but not in the matched primary.

While genome sequencing and expression profiling provides large amounts of data describing biological properties of cancers, these frequently do not reflect genes and pathways that are functionally important for survival and proliferation of tumor cells. Thus, the development of effective targeted therapies will require a better understanding of both the genetic and functional differences within metastatic disease.

In this study, we utilized functional genomic and genomic profiling technologies to perform comprehensive analysis of unique collection of "matched" pairs of HNSCC lines derived from primary tumors and corresponding cervical lymph node metastases. All the cell lines were derived from patients with HPV negative tumors that are known to have worse prognosis compared to the HPV positive subpopulation and will likely require a separate type of treatment for their eradication. Interestingly, both functional and genomic analyses identified a differential and key survival role of the Notch3 signaling pathway in the metastatic lines compared to those derived from primary tumors.

The Notch signaling pathway is conserved from Drosophila to human and plays a central role in development, self-renewal, and differentiation[8–10]. The role of Notch signaling in cancer is being extensively investigated and newly developed drugs that target different arms of the pathway show promising results in preclinical studies[11–15]. Paradoxically, in HNSCC the Notch pathway has been shown to play roles in both oncogenic and tumor suppressor activities. Several reports demonstrate high frequency of loss-of-function mutations in *Notch1* in HNSCC tumors, and consequently a tumor-suppressive role of Notch signaling in at least some subtypes of this disease[16–20]. In fact, close to 19% of HNSCC tumors harbor inactivating mutations in *Notch1*, making it the second most commonly mutated gene after *TP53*. However, other recent reports demonstrate an overexpression of Notch pathway components in HNSCC tumors suggesting oncogenic properties of the pathway[21–25]. Genes with elevated expression included the receptors (*Notch1, Notch2* and *Notch3*), ligands (*Jag1, Jag2*) and target genes *Hes1* and *Hey1*[21–25]. Importantly, functional consequences of pathway activation have also been reported. Pharmacological inhibition of the Notch signaling by gamma-secretase inhibitors or knocking down *Notch1* significantly reduce cell proliferation and invasion of HNSCC cells[14,26]. While most mutational and expression profiling data in HNSCC has been performed on primary tumors, recent analysis

of metastasis-derived samples suggested increased alterations in Notch signaling and reported mutations in Notch3 gene[7]. These observations suggest a dual role of Notch in HNSCC, which is context dependent and needs to be further investigated.

Our findings provide with important insight into HNSCC pathogenesis, suggesting that metastatic cells acquire dependency on Notch3 signaling that may be amenable to targeting metastatic disease.

## Results

**Molecular profiling reveals common changes in matched pairs of HNSCC primary tumor and metastasis-derived cell lines.** Identification of both genetic and functional differences between primary and metastatic variants of HNSCC would be enabled by in vitro models derived independently from these sites from the same patient. Unfortunately, HNSCC tumor cells have typically been difficult to isolate and adapt to grow in vitro and as a consequence there are few commercially available HNSCC cell lines. Consequently, HNSCC is underrepresented in whole-genome functional screens that have been carried out to date[27–30]. However, large programs specifically focused on HNSCC have changed this, including efforts by the Grenman Lab who have established >100 HNSCC lines with associated clinical data[31–34]. Importantly, in many cases multiple lines have been established from the same patient, derived from the primary tumor and a regional metastasis or recurrence. For the purpose of this work, we characterized matched pairs of cell lines derived from primary tumors and corresponding lymph node regional metastases of three HNSCC patients within this collection (Fig. 1A, Table 1, Supplementary data 2). Patients 54, 60 and 74 had primary tumors at different sites (UT-SCC-54A - buccal mucosa, UT-SCC-60A - left tonsil and UT-SCC-74A - tongue) as well as regional lymph node metastasis (UT-SCC-54C, UT-SCC-60B, UT-SCC-74B).

We first conducted exome sequencing on the 6 HNSCC lines from the three patients with the goals of both annotating the known mutations in HNSCC and identifying new mutations uniquely acquired in the metastatic lines. Figure 1B summarizes the mutation profiles of the top 10 most significantly mutated genes in HNSCC[35] across the three matched pairs of cell lines. In all cell lines, HNSCC driver mutations were detected, which included *TP53* mutations in cell lines derived from patients 54 and 74 and *PIK3CA* mutations in cell lines derived from patient 60. *Notch1* loss of function mutations were identified in cell lines derived from patients 54 and 74 along with mutations in *KMT2D*. In each of these cases, mutations were found in both primary and in metastatic lines. However, we also observed acquired mutations that were specific to the cell lines derived from metastatic sites. Within the 10 driver genes, this included acquisition of mutations in *FAT1, KMT2D* and *AJUBA* across the three sets of cell lines. These data demonstrate that ongoing mutational processes, including those that have been identified as driver genes in HNSCC may be uniquely acquired in metastatic variants.

To evaluate the number of mutations unique to the cell lines derived from metastatic sites in more detail, we compared whole exome sequencing between the matched lines, using the primary tumor derived cell lines as a reference. The number of acquired mutations observed in the metastatic lines for each of the three sets is summarized in Fig. 1C. These data demonstrate ~100 unique mutations in UT-SCC-54C and UT-SCC-60B and over 400 new mutations in UT-SCC-74B. To assess changes in pathways that might be common within this set of acquired mutations in the metastatic lines, we carried out gene set enrichment analysis on the newly acquired mutations (for list of

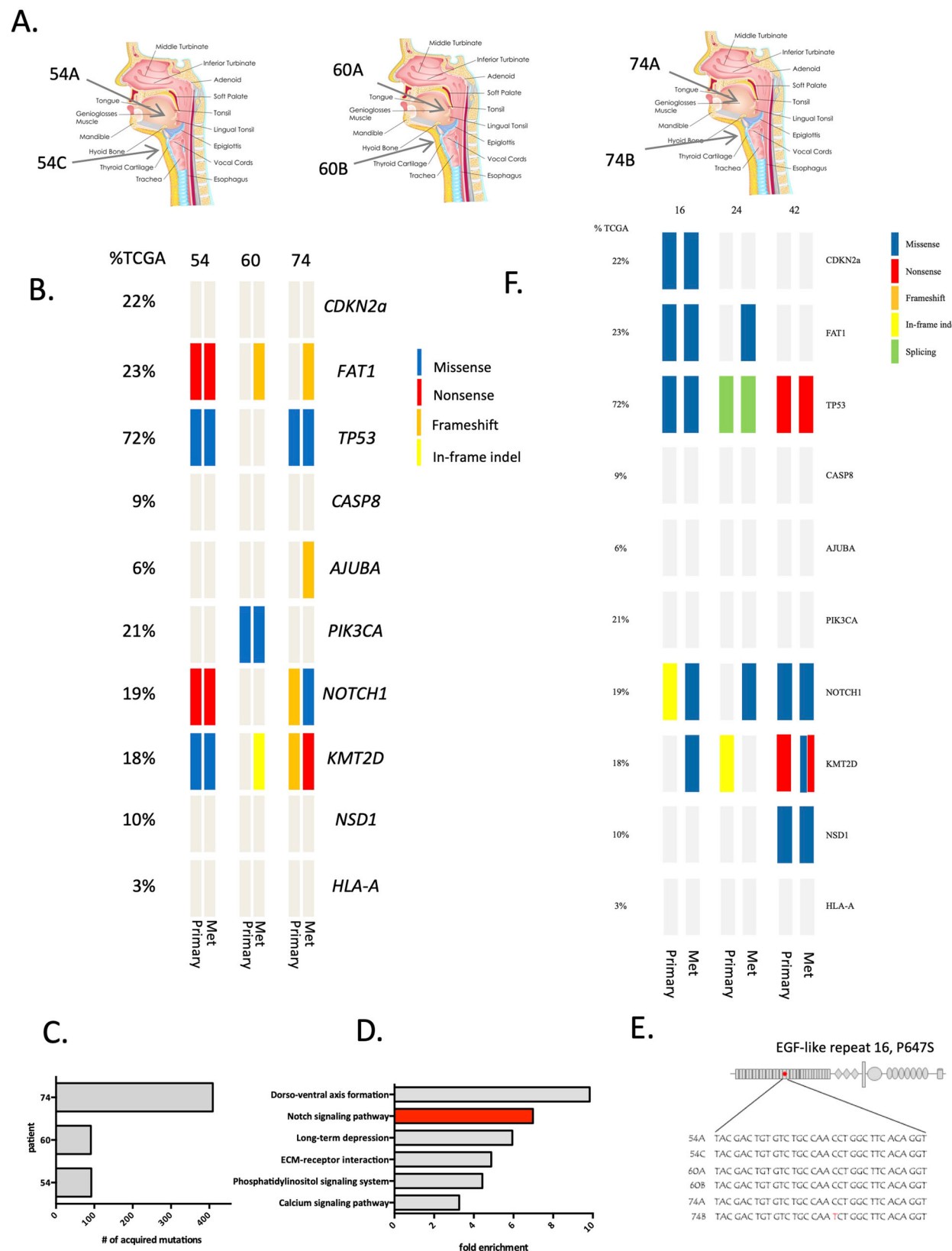

genes used for this analysis see Supplementary data 1). The results revealed 6 pathways with a greater than 3-fold enrichment for being mutated in metastatic cells, including the Notch signaling pathway which was ~7-fold enriched across the three matched samples (p value < 0.05) (Fig. 1D). Other pathways involved in development and differentiation (Dorso-ventral axis formation,

long-term depression), adherence and motility (ECM-receptor interaction) and signal transduction (calcium signaling, phosphatidylinositol phosphate signaling) are worth further investigating as these processes are known to be dysregulated upon metastasis and might serve as unique metastatic therapeutic targets. Moreover, a novel mutation was discovered in Notch3

**Fig. 1 Mutational profiling of matched HNSCC lines reveals a subset of mutations acquired upon metastasis. A** Summary of molecular characteristics and clinical information of the matched pairs of cell lines derived from primary tumors and corresponding metastatic sites of three HNSCC patients (derived and licensed from www.shutterstock.com, original contributor "snapgalleria"); **B** Genomic characterization of the cell lines through exome-based sequencing. Mutational profile of top 10 genes significantly mutated in HNSCC is shown for each line. **C** Number of mutations acquired in metastatic line is shown for each set. **D** Gene-pathway enrichment analysis revealed that Notch pathway was one of the top enriched KEGG pathways among genes uniquely mutated in metastatic cells (*p* value < 0.05). **E** A novel non-synonymous (C to T) acquired mutation in the 16th EGF domain of the NOTCH3 receptor (P647S) was identified in the metastatic (UT-SCC-74B) but not primary (UT-SCC-74A) line. **F** Genomic characterization of the cells lines through targeted sequencing (see Supplementary data 3 for panel of genes sequenced).

**Table. 1 List of resources used for data generation.**

| Reagent or resource | Source | Identifier |
| --- | --- | --- |
| **Antibodies** | | |
| Jag1 | Abcam | Cat# ab7771 RRID: AB_2280547 |
| Jag2 | Abcam | Cat# ab109627 RRID: AB_10860796 |
| Actin | Abcam | Cat# ab8227 RRID: AB_2305186 |
| Hey1 | Novus | Cat# NBP2-47436 |
| Hes1 | Novus | Cat# NBP2-67642 |
| HRP anti-mouse | GE Healthcare | Cat# NA931, RRID: AB_772210 |
| HRP anti-rabbit | GE Healthcare | Cat# GENA934, RRID: AB_2722659 |
| **Chemicals, peptides, and recombinant proteins** | | |
| Human Jagged1 (Ser32-Ser1046) | R&D systems | Accession # P78504 |
| Human Jagged2 (Met27-Leu1082) | R&D systems | Accession #AAB61285 |
| **Deposited data** | | |
| Raw and analyzed data | This paper | GEO: GSE117753 |
| **Experimental Models: Cell Lines** | | |
| UT-SCC-54A | Gift of Dr. Grenman | RRID: CVCL_7863 |
| UT-SCC-54C | Gift of Dr. Grenman | NA |
| UT-SCC-60A | Gift of Dr. Grenman | RRID: CVCL_A089 |
| UT-SCC-60B | Gift of Dr. Grenman | RRID: CVCL_A090 |
| UT-SCC-74A | Gift of Dr. Grenman | NA |
| UT-SCC-74B | Gift of Dr. Grenman | NA |
| **Experimental models: organisms/strains** | | |
| NSG mice | In house colony | IMSR Cat# NM-NSG-001, RRID: IMSR_NM-NSG-001 |
| **Sequence based elements** | | |
| See Supplementary data 10 for list of oligos | | |
| Notch1 smart siRNA pool | Dharmacon | M-007771-02-0005 |
| Notch3 smart siRNA pool | Dharmacon | M-011093-01-0005 |
| Jag1 smart siRNA pool | Dharmacon | M-011060-02-0005 |
| Jag2 smart siRNA pool | Dharmacon | M-017187-00-0005 |
| EG5 smart siRNA pool | Dharmacon | M-003317-01-0005 |
| **Recombinant DNA** | | |
| TRIPZ Notch3 | Dharmacon | RHS4740-EG4854 |
| pLenti CMV Puro Luc w168 | [71] | Addgene #17477 |
| pX458 | [80] | Addgene #48138 |
| H2B-GFP | [81] | Addgene #11680 |
| H2B-RFP | [82] | Addgene #26001 |
| **Software and Algorithms** | | |
| siMEM | [62] | https://doi.org/10.1016/j.cell.2015.11.062 |
| muTect tool | [83] | https://doi.org/10.1038/nature12213 |
| Oncotator | [84] | https://doi.org/10.1002/humu.22771 |
| MutSigCV tool | [83] | https://doi.org/10.1038/nature12213 |

itself in the UT-SCC-74B cell line derived from the metastatic site which was not present in the UT-SCC-74A line derived from the primary tumor (Fig. 1E). This mutation (C to T) is found within the region coding for the 16th EGF-like repeat and results in a non-conservative change from proline to serine at codon 647. The change is predicted to affect protein conformation and to function as a deleterious mutation by Annovar including information from variant effect predictors SIFT, PolyPhen-2, PhyloP, MutationTaster and GERP++[36,37]. Interestingly, a mutation in the 8th EGF domain of *Notch2* was previously discovered in Drosophila and shown to affect the specificity of Notch ligand interaction[38]. EGF repeats in Notch receptors undergo O-fucosylation, to generate functional ligand binding

sites by elongation of polysaccharide chains by Fringe proteins that regulate ligand binding specificity[39,40]. The addition of GlcNac residues by Fringe proteins was shown to inhibit binding of Notch receptors to Delta but enhance binding to Jagged ligands[39]. This suggests a possible functional mechanism by which the newly discovered mutation could affect the metastatic phenotype of UT-SCC-74B cells.

In addition to the whole exome sequencing analysis performed on the 6 cell lines described above, we captured and sequenced the most commonly altered genes in HNSCC as reported by the TCGA in cell lines derived from multiple HNSCC patients (Fig. 1F, Supplementary data 2 and 3). While a comprehensive analysis of this work will be presented in a future publication,

here we report the data from 3 additional patients that had matched lines derived from primary tumors and metastasis. Comparing mutational profiles of these patients between the primary tumor derived cells and their metastatic counterparts, we observed alterations on several genes related to Notch signaling that are present exclusively in metastases. These include Notch receptors Notch1 and Notch4 as well as an EP300 gene that was shown to be associated with Notch signaling pathway[41]. These observations further suggest that genetic alterations in Notch signaling components play a role in the metastatic transformation of HNSCC cells.

Next, we carried out expression profiling across the 6 cell lines to reveal potential common changes in gene expression between the cell lines derived from primary and metastatic sites in each patient. This analysis revealed significant changes in gene expression between primary and metastatic cells including 111 genes which were commonly induced or repressed across the 3 sets of cell lines (Fig. 2a). GO pathway analysis of 37 commonly upregulated genes revealed 17 molecular pathways that were upregulated above 10-fold in metastatic lines (Fig. 2b) while the analysis of 74 commonly downregulated genes revealed 9 pathways that were >10-fold downregulated in metastatic lines across the 3 sets (Fig. 2c). Several pathways involved in cellular differentiation appeared in the upregulated set; interestingly this set also included the Notch signaling pathway that was also found within the enriched mutational data set (Fig. 1D). (For full list of genes and pathways refer to Supplementary data 4). Several pathways involved in cell differentiation and in regulation of adhesion/integrins were upregulated in the metastatic cells. It has been demonstrated by many groups that Notch pathway plays a key role in differentiation which might be correlated with its role in tumorigenesis. For example, inhibition of Notch signaling in hematopoietic progenitor cells (HPC), myeloid-derived suppressor cells (MDSC), and dendritic cells is directly involved in abnormal myeloid cell differentiation in cancer[42]. Activation of Notch signaling in macrophages led to secretion of CCL5, one of the genes found to be upregulated in our metastatic cell collection and led to increased EMT and tumor cell migration[43].

Growing evidence supports a role of Notch interactions with integrins in cancer progression. For example, it has been demonstrated that Notch signaling activates integrin b1 thus enhancing cell adhesion of cancer cells entering blood circulation[44]. Moreover, expression of Jagged-1 was shown to be dependent on several integrin subunits[45]. Overall, the evidence suggests that the cross talk between Notch and integrins is context dependent and is largely regulated by the microenvironment. Thus, it is plausible that a cross talk between Notch3/Jag2 axis and integrins plays a role in HNSCC metastasis which will be explored in further publications.

It is important to keep in mind, that upregulation of Notch signaling in metastasis was not as strikingly significant as for some of the other pathways. The pathway is ranked number 11 amongst the top upregulated and the $p$ value associated with it was 0.07. However, combined with evidence from mutational profiling and functional genomics, the differential expression evidence was an interesting observation that required further evaluation.

Although Notch1 loss of function mutations are commonly found in HNSCC (including in the two cell lines in this study), components of the Notch pathway have also been reported by others to be overexpressed in a subset of HNSCC patients suggesting a potential oncogenic role of Notch signaling in at least some cases of HNSCC. Oncogenic roles for the Notch pathway, and in particular Notch3, have similarly been noted in many other types of cancer[11,12,46–53]. One plausible explanation for this "dual" role of Notch signaling is that different components of the pathway play distinct roles in tumorigenesis. To address this possibility and to validate our gene expression data, we assessed the expression of several Notch pathway ligands and target genes in our cell lines by qPCR and western blot analysis. Interestingly, Jag1 and Jag2 ligands showed opposing expression patterns in primary and metastatic cells (Fig. 2d, e, Table 1 for antibodies used). At the mRNA level, we found that Jag1 is expressed at higher levels in cells derived from primary tumors in all three sets and is significantly downregulated in cell lines derived from metastatic sites. In contrast, Jag2 is significantly elevated in metastatic cells compared to those derived from the primary tumors, and in UT-SCC-54C and UT-SCC-74B (metastatic) cells reach ~4-fold and ~6-fold higher expression than their corresponding primary tumor derived counterparts. Similarly, at the protein level Jag1 is strongly repressed in the cell lines from metastatic sites in all 3 sets and Jag 2 is maintained or upregulated in metastatic lines. Moreover, Hes1 and Hey1, the most known transcription factors regulated by Notch signaling, also showed a consistent differential pattern of expression in cell lines from primary and metastatic sites (Fig. 2e). Hes1 was highly expressed in cell lines derived from the primary tumors and was downregulated in metastatic cell lines whereas Hey1 exhibited an opposite pattern of expression being strongly upregulated in metastatic lines in all three sets. These findings suggest that a distinct cascade of Notch signaling may be constitutively active in the cell lines derived from metastatic sites as compared to those from the primary tumor. Growing evidence suggests that epithelial to mesenchymal transition (EMT) of tumor cells is required for the metastatic progression and acquisition of invasive phenotype[54]. EMT is associated with change in expression of several genes encoding cytoskeletal components of the cells; high expression of vimentin is normally associated with mesenchymal phenotype while E-cadherin is a known marker of epithelial cells. In line with this hypothesis, we observed higher expression of E-cadherin in cell lines derived from primary tumors while vimentin was highly expressed in metastasis-derived cell lines (Fig. 2f).

**Identification of functional differences in matched pairs of HNSCC primary tumor and metastasis-derived cell lines.** Identification of genome wide functional properties has been aided by the development of large lentiviral delivered shRNA libraries targeting the majority of human genes by the RNAi consortium and have been used to discover genes that are essential for the survival of cultured tumor cells from breast, pancreas and ovarian cancers in our institute and elsewhere[55–62]. This approach has also been successfully applied to discover novel synthetic lethal interactions with the RAS oncogene in colorectal cancer cells and to identify dependency between NfκB, FAS and EGFR pathways in lung cancer[60,63].

Given the evidence for common mutational and molecular changes in the cell lines derived from metastatic sites, we hypothesized that the lines derived from metastatic sites may also harbor common unique vulnerabilities that could be identified through functional genomic approaches. We performed functional genomic screens to identify differences in essential genes on the 3 matched pairs of cell lines (Fig. 3a, Table 1 for resources used). Near whole-genome shRNA dropout screens were carried out using the lentiviral 80 K shRNA library[62]. Cell populations were analyzed after ~3 and ~6 population doublings following infection and the siMEM hierarchical regression algorithm was applied to test for differences in gene essentiality between matched pairs of cell lines derived from primary and metastatic sites – see "Methods" for more details[62]. Interestingly, several components of the Notch pathway were identified as differentially essential genes for survival of metastasis-derived cell

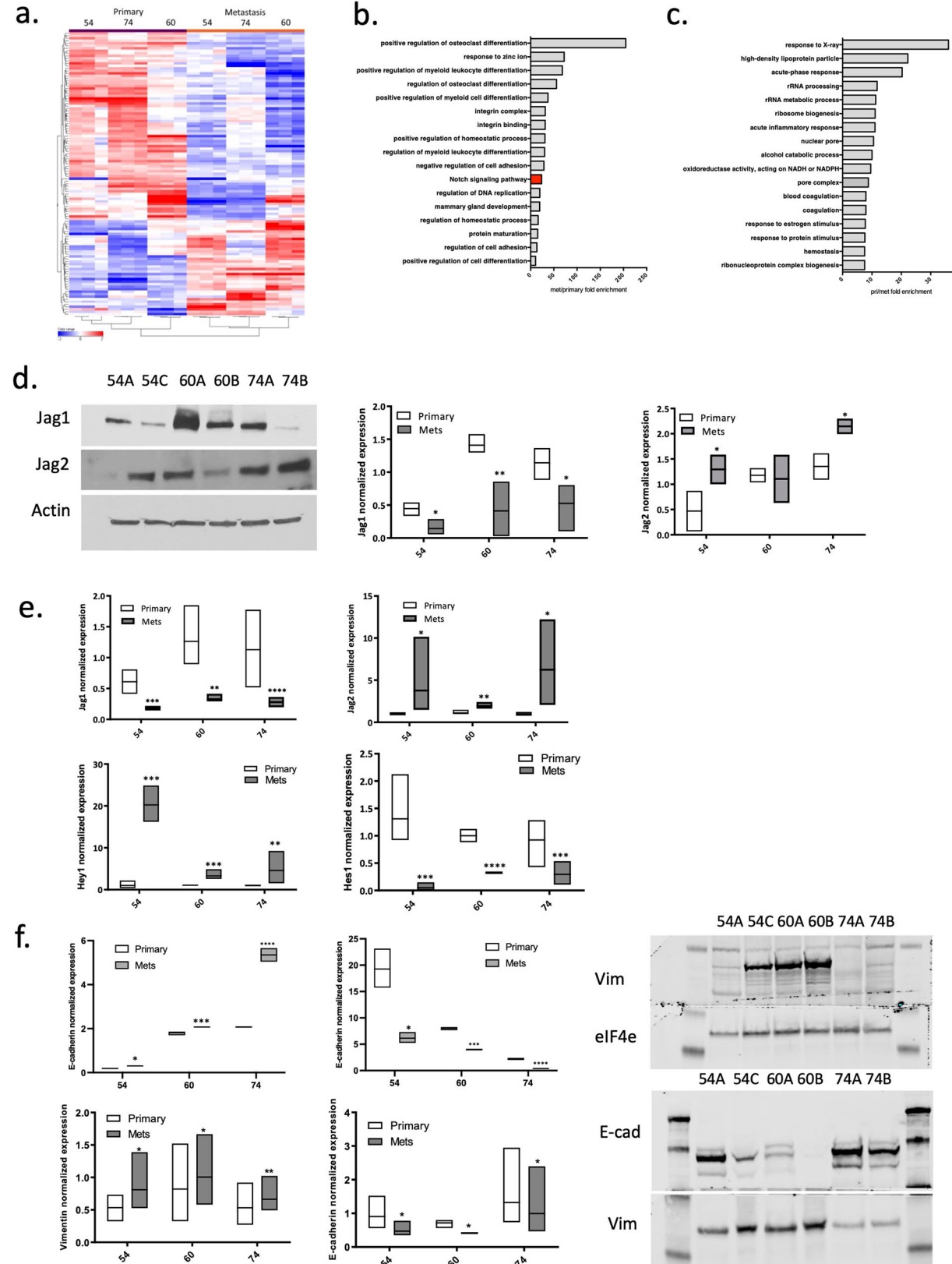

lines (Fig. 3b, b) including *Notch3* itself, its target *HeyL*, the Notch ligand *LCK* which promotes anti-apoptotic effects of Notch1 in T cells[64]. For full list of genes and their essentiality scores refer to Supplementary data 5.

In addition to the 6 cell lines described above, we performed shRNA screens on a larger collection of HNSCC lines developed by Dr. Grenman (Supplementary data 2); a comprehensive analysis of these will be reported in a later publication. For the purpose of current study, we looked at the effects of *Notch3* knockdown across the larger cohort of patients (Fig. 3d). We analyzed *Notch3* essentiality scores in multiple cell lines derived from primary tumors, recurrences and metastatic sites using

**Fig. 2 Differential expression of Notch pathway components is observed in metastatic HNSCC lines. a** Gene expression data. Genes differentially expressed between primary tumor derived and metastasis-derived lines across the 3 sets are shown. **b** GO pathway analysis revealed that several molecular pathways associated with metastatic progression are upregulated in metastasis-derived lines, including the Notch signaling pathway (red). **c** Molecular pathways that are downregulated in metastatic cells are also shown. **d** Notch ligands (Jag1, Jag2) and targets genes (Hes1, Hey1) exhibit different patterns of expression in metastatic and primary tumor lines as established by western blot (**d**) and by qPCR (**e**). **f** qPCR and western blot analysis of EMT markers (Vimentin and E-cadherin) in 3 primary tumor lines and 3 metastatic lines. For the qPCR analysis presented in this figure, the data is normalized to average expression levels in 3 primary tumor cell lines and presented as mean ± SEM.

Expectation Maximization algorithm[65]. The analysis revealed that while cell lines derived from the primary tumors belonged to one population and were not dependent on Notch3, cell lines derived from metastatic sites divided into 2 subpopulations one being strongly dependent on *Notch3* ("low subpopulation") and the another not dependent on *Notch3* ("high subpopulation"). This data confirmed that metastatic cells are differentially dependent on *Notch3* and expanded this observation to a larger cohort of 29 cell lines derived from HNSCC patients.

To validate the targets identified in the shRNA screens, we applied an independent siRNA technology coupled with live imaging of the cells following knockdown (Fig. 3f, Fig. S1). Metastatic and primary lines from each set were labeled by infecting with lentivirus expressing either H2B-GFP or H2B-RFP. Competitive survival/proliferation was assessed by comparing the ratio of primary to metastatic cells as a function of time and compared with both positive (targeting known essential genes) and negative (scrambled siRNA) controls. This analysis confirmed that knocking down *Notch3* selectively killed metastatic but not primary tumor cells. Interestingly, transfection with siRNA against *Jag2*, the ligand showing preferential expression in the metastatic lines, also resulted in selective killing of metastatic cells. In comparison, neither *Notch1* nor *Jag1* showed differential selectivity.

To further validate these results, we engineered doxycycline-inducible shRNA knockdowns of *Notch3* in the three metastatic and three primary lines and studied their growth upon induction of *Notch3* knockdown (Table 1 for resources used). Figure 3f shows that the growth of metastatic lines is inhibited by induction of *Notch3* shRNA while the growth of primary tumor cells is unaffected. This result further confirms our observations from the functional genomics screens and siRNA validation studies and provides a rationale for testing the effect of *Notch3* inducible knockdowns on tumor growth in vivo.

**Differential Notch signaling in matched pairs of HNSCC primary tumor and metastasis-derived cell lines.** While numerous findings support a role of Notch signaling in tumorigenesis, unlike in some other diseases, this effect is rarely mediated by activating mutations in pathway components[12,48,51,66–68]. Our findings demonstrate that in addition to Notch3 dependency, Jag2 was upregulated and differentially essential in metastatic lines while Jag1 was downregulated and non-essential (Figs. 2 and 3). Furthermore, gene expression analysis demonstrated differential Notch downstream signaling in the primary and metastatic lines (Fig. 2f). Differential Notch ligand specificity has been shown to influence cell phenotype in other scenarios[69]. Therefore, it is plausible, that the unique dependency on Jag2 and Notch3 in HNSCC reflects the relative signaling role between specific Notch ligands and receptors within their environment. Such interactions are mediated by binding of Notch receptors to their ligands, all of which are transmembrane proteins expressed by interacting cells[9].

To explore this possibility, we evaluated downstream Notch signaling in response to specific Notch ligands in the matched cell lines. For this purpose, we exposed cells to recombinant ligands immobilized to the plastic surface, which has been shown to properly activate the pathway in contrast to soluble ligands which

can inhibit pathway activity[70], and assessed downstream changes in *Hes1, Hey1,* and in the EMT markers *Vimentin* and *E-Cadherin.* As described above, primary tumor cells have high levels of endogenously expressed Jag1 (Fig. 2d, e), therefore we did not expect that exposure of exogenous Jag1 to these cells would cause significant effect on their phenotype. Instead, we were interested to assess if Jag2 exposure would alter the phenotype of primary tumor cells causing them to become more similar to their metastatic counterparts. Similarly, in the context of metastasis-derived cells we wanted to assess if Jag1 exposure would influence expression of genes in these lines towards a phenotype more similar to cells derived from the primary tumors. Interestingly, metastatic cells cultured on Jag1 expressed higher levels of *Hes1*, a target gene that has high expression in cells derived from the primary tumors (Fig. 4a), while exposure to Jag2 increased expression of *Hey1* in both primary tumor and metastasis-derived cells (Fig. 4b). Moreover, when cells derived from the primary tumors were cultured on Jag2, the expression of vimentin increased indicating that interactions with this ligand can promote EMT of the cells (Fig. 4c). In addition, exposure of metastatic lines to Jag1 decreased expression of vimentin and increased expression of E-cadherin thus inducing an epithelial-like phenotype of the cells (Fig. 4c, d). The latter observation is especially interesting since it indicates that EMT in HNSCC may be reversed, and that that inhibition of Notch3-Jag2 signaling in these cells might have therapeutic potential.

**Differential expression of Notch pathway components in TMA constructed from HNSCC patient tumors.** While HNSCC derived cell lines used in this study represent a valuable tool to study this disease, their biology might be affected by passaging in culture and might not always accurately represent the biology of the tumors they were derived from. Therefore, we decided to look at the expression of Notch pathway components in a patient material from HNSCC surgical samples constructed into a TMA (Supplementary data 6). Cores that had obvious artifacts due to edge effects, air bubbles, etc. were eliminated from the analysis. Regions that contained non-tumor tissue such as normal salivary gland or muscle were also excluded (Fig. S2).

Using immunohistochemistry, we looked at the expression of Jag1, Jag2, Hes1 and Hey1 in the TMA. We determined that for each one of proteins the cores can be divided into "low" "intermediate" and "high" populations based on the percentage of positive cells within the sample; Jag1 only had a "high" and "low" populations (Fig. 5a). Interestingly, higher proportion of patients with late stages of HNSCC (T3 and T4) belonged to the "high" Jagged2 and Hey1 populations as compared to patients with low stages of HNSCC that mostly expressed low or intermediate levels of these proteins (Fig. 5b, Supplementary data 7). No significant correlation between tumor stage and the expression of Hes1 and Jagged1 or Notch3 proteins was observed (Fig. S3). This data is in line with our observations that Jagged2 and Hey1 expression is elevated in cells derived from metastatic sites and is low in cells derived from primary tumors. Interestingly, patients with high/intermediate proportion of Jagged2 positive cells also had a high proportion of cells expressing Hey1, confirming our hypothesis

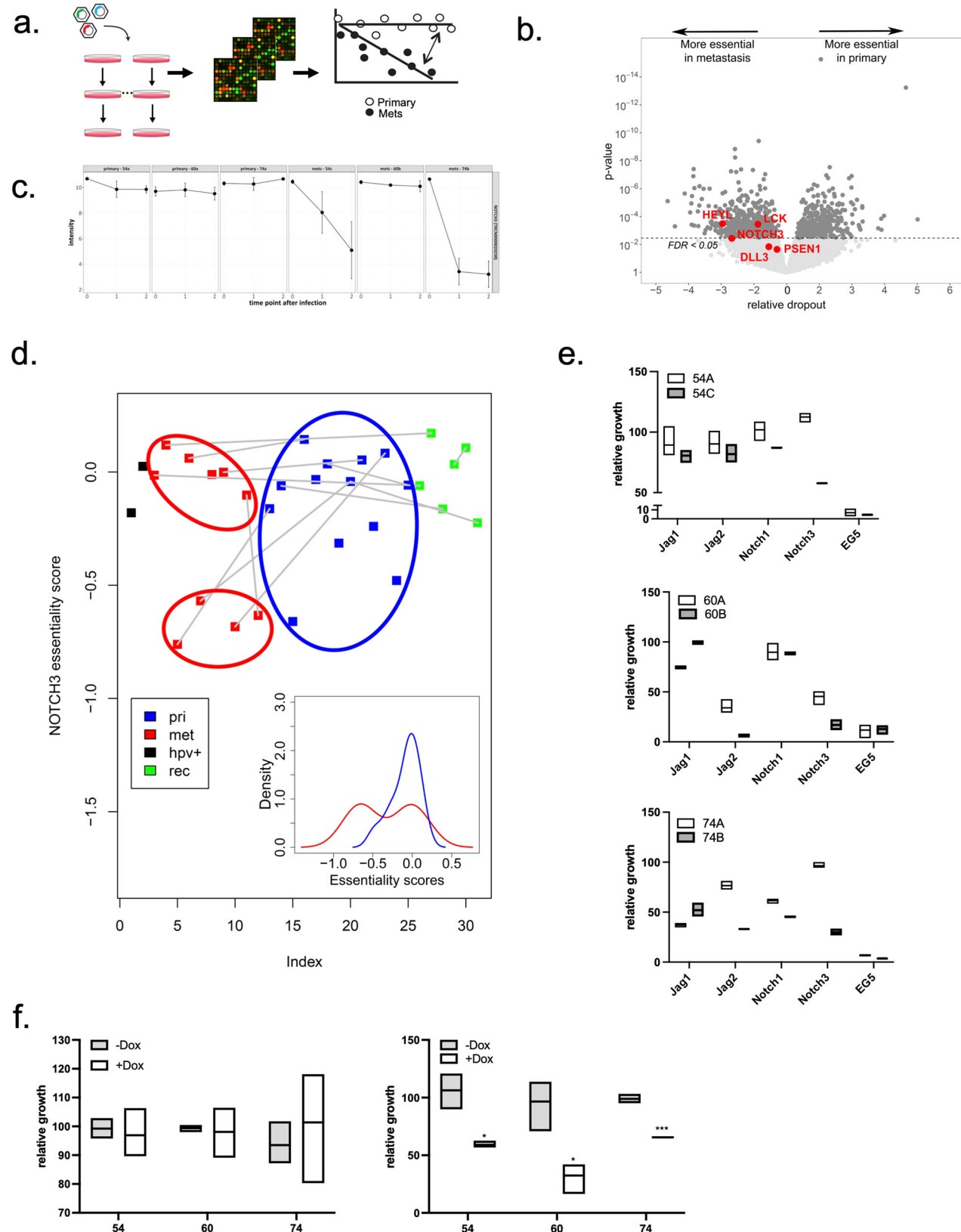

that these 2 proteins belong to the same signaling cascade that is activated during metastatic progression (Fig. 5c). Crosstab analysis of all available clinical information is shown in Supplementary data 8, univariant and multivariant analysis of correlation between protein expression and OS/DFS are shown in Supplementary data 9. Overall and disease-free survival analysis based on

expression of Notch pathway components is shown in Fig. S4. We did not observe a correlation between survival and the expression of any of the markers tested; expression of those proteins is indicative of a current state of a tumor rather than being prognosis predictive. This is in line with the findings illustrated in Figs. 2 and 4, that Jag1/Jag2 and Hes1/Hey1 are differentially

**Fig. 3 Pooled screens reveal metastasis-specific vulnerabilities. a** The analysis workflow of pooled shRNA screens[61] in primary and metastatic cells. The screens are quantified by microarrays, after which all measurements are extracted, and each measurement is annotated with gene, shRNA, cell line, time point, replicate and primary/metastasis information. The siMEM hierarchical regression algorithm[62] is then applied to test for differences between primary and metastasis cells. **b** Volcano plot illustrating significant predictions from the primary vs. metastasis analysis, including several components of the Notch pathway (red). **c** The abundance of NOTCH3 shRNA is reflected by the fluorescence intensity in the Affymetrix array measured across the 3 time points of the screen (0, 1 and 2, corresponding to T0, T1 and T2, respectively). For each patient there are 2 plots, first showing the result from the primary tumor sample and the second showing the result from the metastatic sample from the same patient. Each time point in each of the plots is represented by 3 replicates. **d** Notch3 essentiality scores across a set of 29 cell lines (see Supplementary data 5 for details) are shown. The x axis (labeled index) represents the ordered list of the cell lines shown in the plot. Samples are paired based on the patient they were obtained from, and each pair is connected by a gray line. Cell lines derived from the primary tumors are contoured with a blue line; cell lines derived from the metastatic samples are contoured with red lines. "Pri" primary tumor derived cell lines, "met" metastases derived cell lines, "HPV+" HPV positive cell lines, "rec" recurrence derived cell lines. Insert: Distributions of Notch3 essentiality scores of primary tumors derived cell lines (blue) and metastases derived cell lines (red) are shown on a density plot that shows frequency of each score in each cell line. For cell lines derived from metastatic samples, the distribution of essentiality scores is bimodal, allowing to separate the cell lines into 2 groups (see red contours). Comparison between these 2 groups showed a significant difference (Welch Two Sample t-test, $p = 4.029e-05$); additionally, the bottom group of metastatic samples was significantly different from the primary tumors (Welch Two Sample t-test, $p = 1.555e-06$). **e** Metastatic and primary lines from each set were labeled by infecting with lentivirus expressing either H2B-GFP or H2B-RFP, mixed in an equal ratio (300 cells/well of each line), plated in 384 well plates, and transfected with Dharmacon siRNA pools (using RNAmax Lipofectamine) against the target of interest. Fluorescent images were acquired using In Cell analyzer and shown in Fig. S1. Competitive survival/proliferation is assessed by comparing the ratio of primary/metastatic cells as a function of time. The data is presented as mean ± SEM. **f** HNSCC lines were engineered to express (doxycycline) inducible shRNA against Notch3. Cellular growth upon induction of the shRNA against Notch3 was measured in primary and metastatic lines using Incucyte Zoom Instrument and the increase in confluency between day 5 and day 1 was calculated and normalized to control (untreated) cells for each cell line. The graph illustrates that knockdown of Notch3 selectively inhibits growth of metastatic cells. The data is presented as mean ± SEM.

---

expressed in cells that are already of primary tumor/metastasis state rather than predictive of metastatic potential. Together with the differential dependency observation, these findings are of clinical relevancy as they suggest that Jag2/Notch3/Hey1 axis can be used for development of novel treatments for metastatic HNSCC.

**A Notch3 mutation acquired in metastatic tumor cells contributes to the metastatic phenotype.** As reported in Fig. 1, whole exome sequencing of the "matched" pairs of lines revealed a novel mutation in one of the extracellular EGF domains of the Notch3 receptor that was predicted to be deleterious. This mutation was acquired in the metastatic line, and not present at detectable levels in the primary cell line. To test whether the described mutation has a functional role, we utilized the CRISPR/Cas9 system to 'correct' the mutation in UT-SCC-74B cells, reverting them to the wild-type genotype at this locus. For this purpose, cells were co-transfected with a plasmid that encodes both the *CAS9* gene and the sgRNA for *Notch3* and with a pair of oligonucleotides serving as a repair template. The repair event was validated using a custom designed TaqMan® SNP Genotyping Assay (Life Technologies Corporation) (Fig. S5).

Interestingly, *Notch3* corrected UT-SCC-74B cells exhibited a slower growth rate as compared to the original UT-SCC-74B cells (*p* value < 0.0001, Fig. 6a) and resembled the UT-SCC-74A cell line derived from the primary tumor. Similarly, the UT-SCC-74B corrected line demonstrated reversion of the altered expression patterns of *Hes1* and *Hey1* target genes and the Jag2 Notch ligand (Fig. 6b–d). Moreover, the CRISPR/Cas9 corrected UT-SCC-74B cell line showed decreased expression of *vimentin* and increased expression of *E-cadherin* consistent with a reversion back to a more epithelial phenotype consistent with the primary derived cell line. Most importantly, correction of the *Notch3* mutation conferred loss of dependency on Notch3 for survival as assessed by siRNA knockdown (Fig. 6e), indicating that the mutation in *Notch3* was directly responsible for this dependency.

**Inhibition of Notch signaling has a differential effect on tumor growth and survival in mice bearing subcutaneous tumors seeded by metastatic cell lines.** Our data suggest that inhibition

of Notch3 signaling, and in particular, the Jag2/Notch3/Hey1 axis may be an effective therapy for HNSCC metastatic disease. As a first step to explore this in a therapeutic context we evaluated the consequence of genetic inhibition of Notch3 signaling in vivo in mice. We engineered UT-SCC-74A and UT-SCC-74B cells to express a shRNA against *Notch3* under the control of doxycycline and injected the cells subcutaneously into NOD/SCID mice[71]. When tumors reached a volume of ~200mm$^3$, the mice were divided into control and (doxycycline) treated groups and tumor growth was monitored weekly by caliper. We discovered that the knockdown of Notch3 led to a significantly slower growth of tumors seeded by UT-SCC-74B cells as well as to an improved survival of the mice bearing such tumors (Fig. 7a, b). Interestingly, there was no effect of Notch3 knockdown on the growth of UT-SCC-74A seeded tumors nor was the survival of the mice affected. These data confirmed that while Notch3 expression/signaling is not essential in cells derived from primary HNSCC tumors, it is required for survival and growth of cells derived from HNSCC metastasis. Confirmation of knockdown efficiency and expression of Notch pathway components and EMT markers in tumors are shown in Fig. S6a, b.

**Inhibition of Notch signaling improves survival and inhibits growth of lymph node metastasis in an orthotopic mouse model of HNSCC.** Next, we looked at the effects of Notch3 knockdown in a metastatic HNSCC model seeded from orthotopic implantation. We utilized a previously described approach that involved orthotopic submucosal implantation of the HNSCC cell lines into the tongue, which spontaneously metastasize to the regional lymph nodes in a pattern similar to that in patients[72]. UT-SCC-74B cells transduced with the shRNA against *Notch3* were also transduced with luciferase to allow in vivo imaging and quantification of the tumors. We then injected the cells into the tongues of the mice. When xenografts reached a size of 5–6 mm, they were surgically removed to allow growth of naturally seeded lymph node metastasis. Using bioluminescence, the establishment and growth of the metastatic lesions were monitored (Fig. 8a, b and Fig. S6c). Administration of doxycycline to reduce Notch3 expression significantly inhibited the growth of lymph node metastasis and correspondingly improved survival in mice (Fig. 8c).

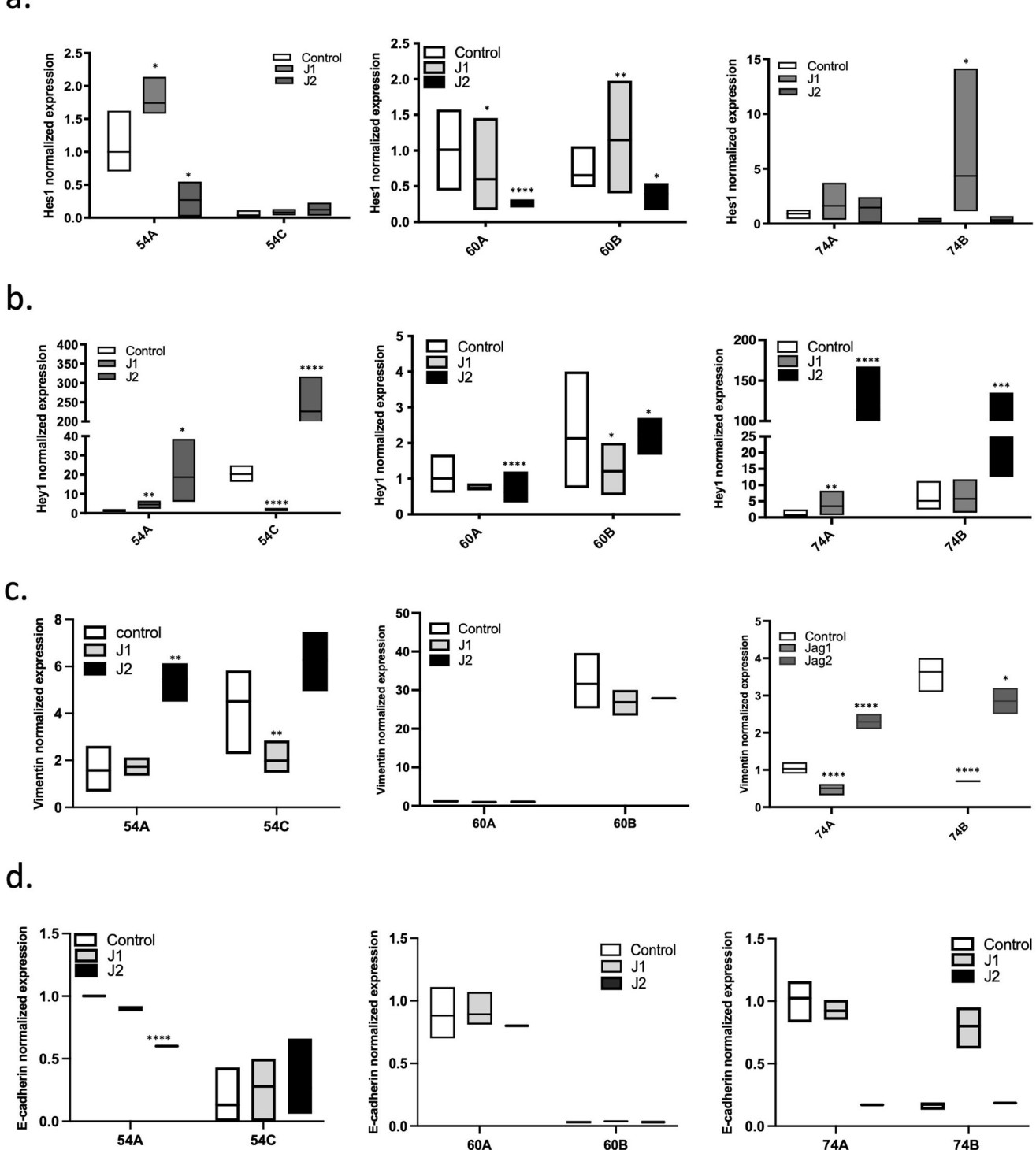

**Fig. 4 Differential Notch ligand specificity determines metastatic phenotype of the cells and leads to activation of distinct target genes. a** Primary tumor and metastatic cells were cultured for 72 h on 6-well plates pre-coated with immobilized Jag1 or Jag2 ligands. Expression of Notch target genes (**a**, **b**) and EMT markers (**c**, **d**) was evaluated in these cells by qPCR. All qPCR data is normalized to HPRT expression and to the average expression levels of each gene in the corresponding primary tumor cell line. The data is presented as mean ± SEM.

qPCR performed on RNA isolated from lymph nodes of treated and control mice revealed significant reduction in *Notch3* expression in mice that received doxycycline compared to control (Fig. S6c). In this experiment, we only utilized UT-SCC-74B cell line derived from HNSCC metastasis and did not look at the corresponding line derived from the primary tumor (UT-SCC-74A). The reason for that is that we aimed to look at the therapeutic potential of Notch3 inhibition in the context of existing metastasis, rather than trying to prevent metastatic transformation. Thus, looking at UT-SCC-74A cells would be therapeutically irrelevant in this model. Taken together, our data indicate that Notch3 is an essential gene for survival and proliferation of metastatic HNSCC cells in vitro and in vivo and targeting this pathway has therapeutic potential in patients with metastatic disease.

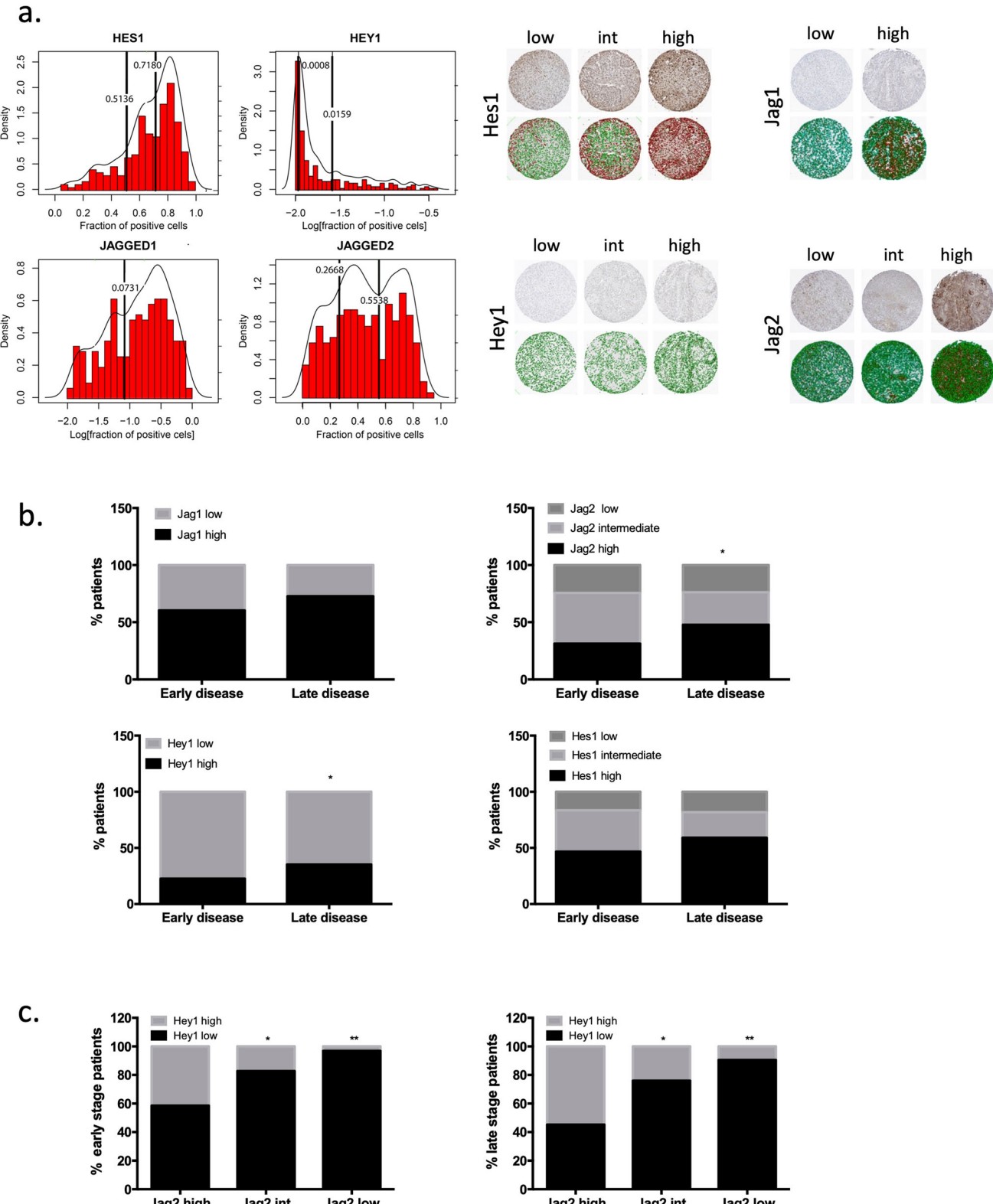

## Discussion

While there have been significant improvements in selective targeting of many malignancies based on molecular diagnostics and targeted agents, cancer progression to metastatic stages remains unmanageable in most cases and the majority of patients die of metastatic disease. Large efforts to understand and target metastasis have concentrated on the molecular characteristics of tumor cells that are involved in metastatic progression[73]. Working with a large collection of HNSCC cell lines derived from primary tumors and "matched" metastatic sites we aimed to identify vulnerabilities that are specific to metastatic cells and that can be exploited to inhibit growth of the existing metastasis rather than preventing metastatic progression to occur. The approach is clearly preferable from a therapeutic standpoint as most mortalities occur in patients with metastases already present at the time of treatment. Our

**Fig. 5 Differential expression of Notch pathway components in TMA constructed from HNSCC patient tumors.** 4 µM sections of the TMA were stained with antibodies against the components of the Notch pathway previously identified as differentially expressed by metastatic HNSCC lines. **a** Distributions of fractions of positive cells for Jag1, Jag2, Hes1 and Hey1 across the TMA cores. Y-axis on the left axis indicates Density values, related to the density curve (the density curve is showing the smoothed distribution of the points along the x-axis). Y-axis on the right indicates frequency values related to the histogram. Thresholds between the subpopulations obtained by using EM algorithm (see "Methods") are indicated with vertical black lines; the exact values of the thresholds are indicated near the corresponding lines. **b** Pearson's Chi-squared tests were performed to examine the association between the pTstages (early and late) and the level of protein expression. Plots for Jagged1, Hes1 are reporting the $p$ values for the general comparison (2 stages vs. 3 categories), the plot for Jagged2 reports the corrected $p$ value for the specific comparison (intermediate/high vs. 2 stages) and the plot for Hey1 reports the $p$ value for the comparison of high/low vs. 2 stages. **c** Proportion of cores that belong to high/low populations of Hey1 is calculated in each population of Jag2. The proportion of high Hey1 cores is higher in high Jag2 populations.

data demonstrate an important role for Notch3 signaling in metastatic variants of HNSCC.

One possible explanation for the dual role of Notch signaling in HNSCC is that different Notch receptors, ligands, or signaling components might play opposing roles in tumorigenesis[74–76]. Our data indicate that while metastatic HNSCC cells show an acquired dependency on Notch3, they are not similarly dependent on Notch1 (Fig. 3e). Interestingly, knockdown of *Notch3* was previously shown to inhibit sphere formation and increase sensitivity to cisplatin in nasopharyngeal carcinoma lines[77]. Moreover, we observed a pattern of increased mRNA and protein expression of *Hey1* in metastatic cells compared to primary tumor cells and a corresponding decrease in *Hes1* expression in cell lines derived from metastatic sites (Fig. 2f) This suggests that *Hes* and *Hey* genes have distinct roles in tumorigenesis and that a 'switch' from *Hes1* to *Hey1* expression is associated with the metastatic phenotype. Moreover, the distinct pattern of *Hes1* and *Hey1* expression correlates with differential expression of Jag1 and Jag2 ligands in both the cell lines and the patient material (Figs. 2 and 5). Taken together, these observations suggest that while Notch1 clearly can act as a tumor suppressor in HNSCC, Notch3 has an oncogenic role, is involved in a distinct signaling cascade and contributes to the survival of metastatic derivatives of tumor cells.

While the differential dependency of metastatic HNSCC cells on Notch3 was clearly evident from the in vitro data, it was important to look at the effects of Notch3 knockdown in a therapeutically relevant context. Thus, we generated subcutaneous tumors from both the metastatic and the primary tumor derived cells and looked at the effect of Notch3 knockdown in both cases. Interestingly, and in line with our hypothesis, tumors seeded by the metastatic cells shrank while no effect was observed on the tumors seeded by primary tumor cells. Thus, cells derived from HNSCC metastasis require Notch3 for their growth and survival at a subcutaneous site in mice, further confirming that the cells acquire Notch3 dependency upon metastatic transformation.

Orthotropic xenograft models are challenging to establish but present with significant advantages compare to subcutaneous models, including a closer correlation to human tumors in their molecular characteristics and metastatic behavior[78,79]. We utilized an orthotropic model for HNSCC xenografts and developed a platform that allows reliable quantification of metastatic growth and response to therapy. Using this model, we demonstrated that inhibition of Notch signaling results in growth inhibition of HNSCC lymph node metastases and improvement in survival. This finding provides a rationale to develop novel, Notch-based targeted therapies for metastatic HNSCC.

In summary, we developed a new approach to treat metastatic HNSCC. This approach is based on uncovering vulnerabilities acquired by cells upon metastatic transformation and targeting these vulnerabilities to kill metastatic cells present at the time of diagnosis. Since most HNSCC patients die of metastatic disease,

this research will lead to development of effective targeted treatments that result in durable cures and can potentially be applied in other metastatic cancers.

## Methods

**Contact for reagent and resource sharing**. Further information regarding materials and methods used in this study may be directed to the co-corresponding author Maria Kondratyev.

**Cell lines**. HNSCC patient-derived lines were generously provided by Dr. Grenman's lab, authenticated by STR and tested for mycoplasma and bacterial contamination. All cell lines were grown at 5% $CO_2$ at 37 °C in Modified Eagle Medium F15 (MEM-F15) supplemented with 10% v/v fetal bovine serum (Gibco, Burlington, Canada) and 1% v/v Fungizone (Gibco).

**Mice**. All animal experimentation was conducted in accordance with the Canadian Guide for the Care and Use of Laboratory Animals, and protocols were approved by the Animal Care Committee at the Princess Margaret Hospital Cancer Center, University Health Network, University of Toronto (Toronto, ON, Canada). NOD-Prkdc$^{scid}$Il2rg$^{em1Smoc}$ Mus musculus (IMSR Cat# NM-NSG-001, RRID: IMSR_NM-NSG-001) were bred in house and used for generation of tongue tumors as described.

**Exome sequencing**. Agilent SureSelect Human All Exon V4 in-solution hybrid was used to capture kit to target genomic DNA region. Captured libraries are subsequently sequenced on an Illumina NextSeq aiming for ~250X coverage. Raw reads were generated according to Illumina NextSeq protocols, generating 100 bp paired end reads and subsequently aligned to the human genome reference sequence build hg19/GRCh37 using BWA followed by realignment around InDels for quality score recalibration. Single Nucleotide Variations (SNVs) were called using muTect tool (https://doi.org/10.1038/nature12213). To obtain metastatic-specific somatic mutation calls, we performed mutation calling for metastatic cell lines, using primary samples as matched normal in Mutect run. We then used Oncotator (10.1002/humu.22771) to annotate the somatic mutations from Mutect. Finally, we performed additional filtering (using Oncotator's annotations for Exome Sequencing Project and dbSNP databases) to further filter germline mutations. Following somatic mutation calling and filtering, we ran MutSigCV tool (https://doi.org/10.1038/nature12213) to determine the acquired mutations. Subsequently, the top ranked genes (196 genes; $p$ value < 0.05), were selected and submitted to The Database for Annotation, Visualization and Integrated Discovery (DAVID) for gene-pathway enrichment analysis.

**Targeted sequencing**. Raw reads were generated according to Illumina NextSeq protocols, generating [100 bp paired end reads] and subsequently aligned to the human genome reference sequence build hg19/GRCh37 using BWA followed by realignment around InDels for quality score recalibration. Single Nucleotide Variations (SNVs) and InDels were called by VarDictJava (v1.7.0) and variant annotation was completed using Annovar (v20180416). Variant calls were filtered to review variants that were exonic, not synonymous, of good quality and at least 1% frequency. Additionally, only those whose gnomAD (v2.1.1) max frequency was less than 0.01 were considered.

**Gene expression profiling**. Global gene expression profiling of the 3 sets of lines was performed using Illumina HT-12 v4 BeadChip arrays within the Princess Margaret Genome Center (PMGC). The data was considered with patient site as conditions and a one-way ANOVA with a Benjamini-Hochberg FDR corrected $p < 0.05$ and post-hoc test (Tukey HSD test) for specific comparison differences was used. The significant probes were then filtered to find only those that were at least 1.5-fold differentially regulated in cell lines derived from all 3 patients primary vs. metastasis. A Benjamini and Yekutieli corrected ($p < 0.1$) hypergeometric test, to look for enriched Gene Ontology categories that may overlap, was applied to each of these lists in turn.

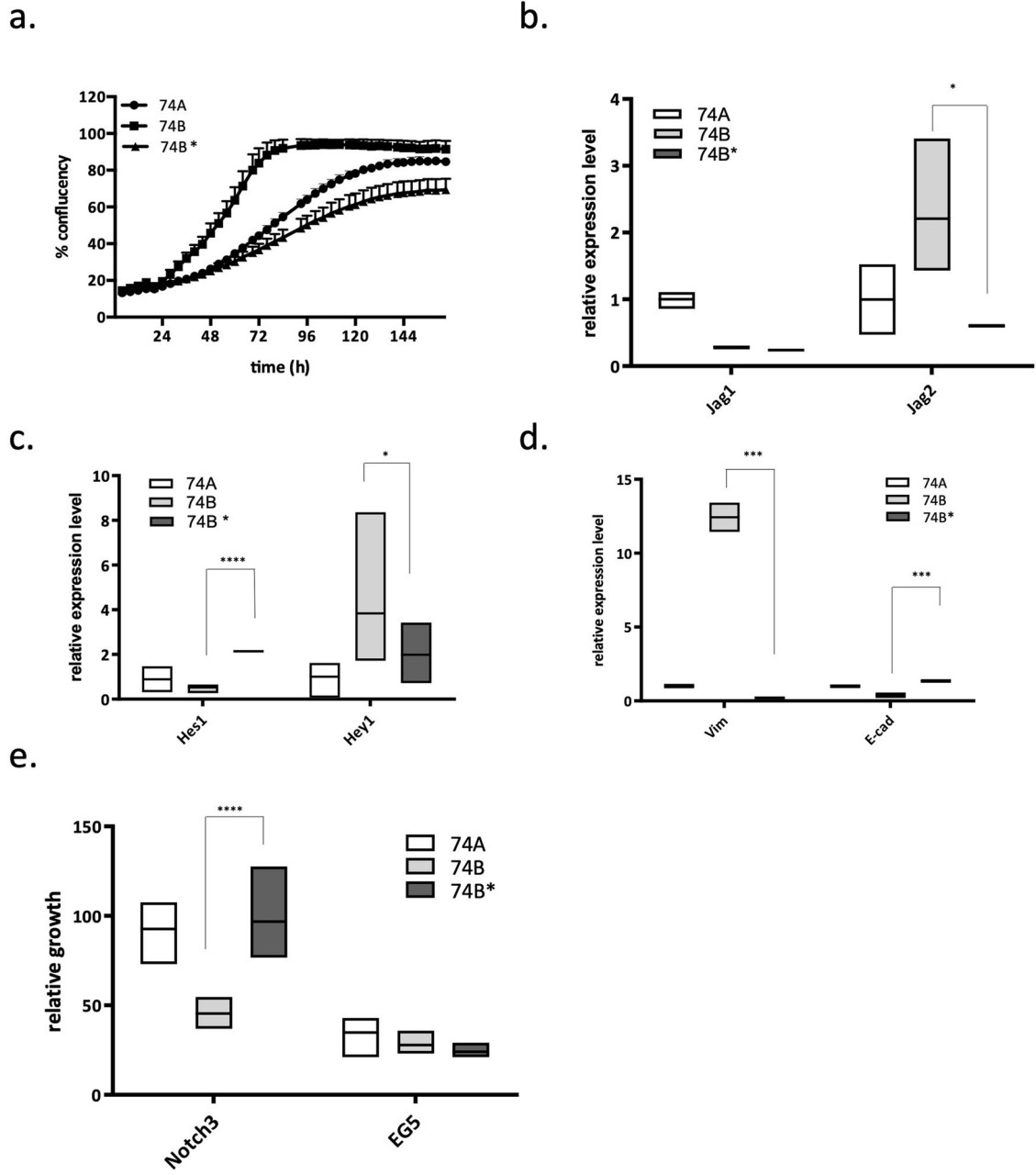

**Fig. 6 A novel acquired mutation in the 16th EGF domain of NOTCH3 contributes to metastatic phenotype of HNSCC cells.** We utilized CRISPR/Cas9 methodology and performed genome editing of UT-SCC-74B line, correcting the point mutation identified in Notch3 gene. CRISPR corrected cell line is labeled as 74B* in all figures. **a** An increase in confluency in parental lines and CRISPR edited clone was accessed over time using Incucyte Zoom Instrument. The growth rate of 74B* line differs from UT-SCC-74B ($p$ value < 0.0001) and resembles the growth rate of UT-SCC-74A. **b–d** The expression of Notch ligands, target genes and markers of EMT in CRISPR edited cell line was accessed by qPCR comparing to parental UT-SCC-74A and UT-SCC-74B cells. **e** Cells were plated in 96-well plates and transfected with siRNA against Notch3. Confluency of cells before and 5 days after transfection was accessed using Incucyte Zoom Instrument; the increase in confluency was calculated and normalized to control (mock transfected) for each cell line. siRNA for EG5 was used as a positive control for transfection. All the data in this figure is presented as mean ± SEM.

**shRNA whole-genome screens**. Near whole-genome shRNA dropout screens were carried out using the lentiviral 80 K shRNA library[62]. Each gene was targeted by average of 5 different shRNAs. Cell populations were analyzed after ~3 and ~6 population doublings following infection and the siMEM hierarchical regression algorithm which models the level of each shRNA as a regression function of the initial abundance of this shRNA (population at time T0), baseline trend in this shRNA's abundance across time points and difference in abundance trend between samples sharing a common feature. In short, this model represents the time course as a line with an intercept and a slope[62]; the slope reflecting the rate of the dropout of the shRNA and termed "essentiality score". The siMEM algorithm was also applied to test for differences in gene essentiality between matched pairs of cell

lines derived from primary and metastatic sites which yielded p-values indicating statistical significance of these differences.

**siRNA validation studies**. Metastatic and primary lines derived from each patient were labeled by infecting with lentivirus expressing either H2B-GFP or H2B-RFP respectively. Nuclear labeling facilitated accurate quantification through repeated live-cell imaging and quantification of fluorescent cells (In Cell Analyser). Using robots, the lines were mixed in an equal ratio (300 cells/well of each line), plated in 384 well plates, and transfected with siRNA against the target of interest (Dharmacon smart pool siRNA were used, see resource table for details). Competitive

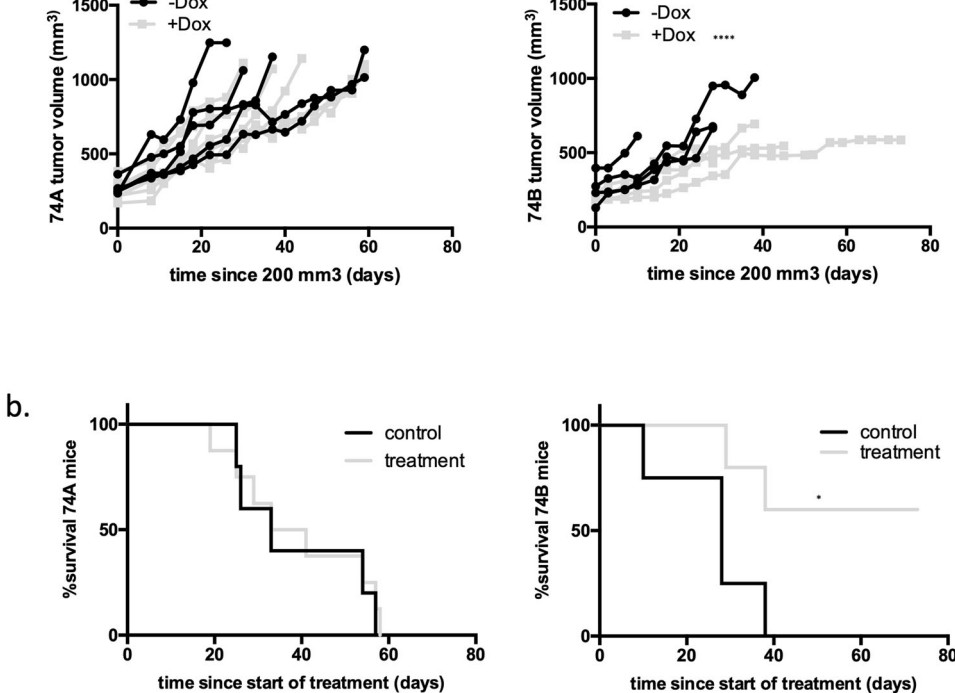

**Fig. 7 Inhibition of Notch signaling has a differential effect on tumor growth and survival in mice bearing subcutaneous tumors seeded by metastatic cell lines. a** Mice were injected subcutaneously with 2 M cells from UT-SCC-74A and UT-SCC-74B transduced with inducible shRNA for Notch3. When tumors reached 200 mm³ the mice were separated into control and treatment groups; the later received doxycycline in their food. The tumors were measured weekly using caliper until mice reached experimental endpoint. **b** Survival analysis of control and doxycycline treated mice.

survival/proliferation is assessed by comparing the ratio of primary: metastatic cells as a function of time and compared with both positive (targeting known essential genes) and negative (scrambled siRNA) controls.

**TMA - Image acquisition and analysis**. Each TMA was imaged at 40x magnification using an Aperio ScanScope AT2. The digitized images, each containing approximately 200 individual cores, were loaded into Visiopharm software (version 2019.10, Visiopharm, Hoersholm, Denmark), in which each core was isolated and analyzed separately. The nuclear counterstain was used to identify a cell nucleus, and a cytoplasm was simulated. To visualize the segmented cores, an overlay showing nuclear segmentation (light blue), negative cytoplasm (green) and positive cytoplasm (red) is shown in Fig. 5 below a screenshot of the original core image. Statistical outputs relating to the number of cells positive for particular markers, and the brown intensity within those cells, were exported for subsequent statistical analysis.

**TMA—filtering of cores**. Selection of TMA samples was based on a 3-step procedure. During Step 1, distribution of artifact area fractions was analyzed across the 445 samples. Mixed Gaussian model was fitted by using EM algorithm, and the lowest Gaussian was selected to represent the population with low artifact area fractions, leading to a threshold of 0.07. The remaining subset contained 288 samples, and the distribution of their tumor area fractions was analyzed to find a threshold for selecting samples with high tumor area fractions (Step 2). Mixed Gaussian model was fitted by using EM and the lowest Gaussian was selected to represent the population with low tumor area fractions, leading to a threshold of 0.3. Remaining 231 samples had tumor area fractions higher than 0.3, and, therefore, were defined as samples with high tumor area fractions. Step 3 consisted of examining the ratio between tumor area fraction and artifact area fraction for each of the 231 samples. The goal of the selection was to find samples with low artifact area fraction and high tumor area fraction, which was higher than the artifact area fraction. Out of the 231 samples obtained in Step 2, all samples had tumor area fraction > artifact area fraction; therefore, the final cohort contained all of the 231 samples obtained in Step 2. The filtering procedure is illustrated in Fig. S2.

**TMA—threshold calculation**. For Hey1, Hes1 and Jagged2, the optimal number of Gaussians fitted by EM algorithm for the majority of the distributions resulted

from 1000 iterations of the 90% resampling was more than 2 Gaussian populations (654, 1000 and 850, respectively), while for Jagged1, the optimal number of Gaussians was 2 for 982 out of the 1000 iteration of the same resampling approach. Therefore, we used 2 thresholds for the Hey1, Hes1 and Jagged2 sample populations and 1 threshold for the Jagged1 sample population. To define the thresholds for each of the proteins, we calculated means for each of the thresholds obtained from the iterations of resampling yielding more than 2 Gaussians for Hey1, Hes1 and Jagged2, and from the iterations of resampling yielding 2 Gaussians for Jagged2. Following this procedure, Hey1, Hes1 and Jagged2 exhibited 3 categories of expression: (1) high, (2) intermediate, and (3) low, while Jagged1 exhibited 2 categories of expression: (1) high, and (2) low.

**TMA—pT stages comparisons**. pTstages were stratified into 2 categories: early (pTstages 1–2) and late (pTstages 3–4). To examine the association between the pTstages (early and late) and the level of protein expression (high, intermediate, and low), we performed Pearson's Chi-squared tests for each of the 4 proteins. Since the $p$ value obtained from the Pearson's Chi-squared tests performed on the data obtained for Jagged2 was significant ($p$ value = 0.02662), Pearson's Chi-squared tests with simulated $p$ value (based on 2000 replicates). Obtained $p$ values were corrected for multiple testing with BH correction; adjusted $p$ values <0.05 were considered to be significant.

**Orthotopic mouse model**. A total of 0.5 × 10⁶ cells of UT-SCC-74B cells transduced with luciferase expressing virus (pLenti CMV Puro LUC w168 Addgene plasmid # 17477) and with the inducible shRNA for Notch 3 (TRIPZ Notch3, RHS4740-EG4854, Dharmacon) were resuspended in 30ul FBS-free Eagle's Minimum Essential Media (Sigma-Aldrich M8042-6X500ML) and injected into the tip of the tongue of female 8–12-week-old NSG mice using a 30G needle. Mice were given isoflurane inhalant as an anesthetic; 5% was used for induction, and 2% for maintenance thereafter. Mice were imaged using Xenogen IVIS-200 instrument weekly after injection. When xenografts reached a size of 5–6 mm, they were surgically removed to allow growth of naturally seeded lymph node metastasis. For this purpose, mice were given 5 mg/kg SQ meloxicam (CDMV, cat# 104674) as analgesic prior to surgery and once a day for two days after surgery for pain management. Once the animals reached surgical plane, as confirmed by toe pinch and respiration rate, the tongue tumor was grasped by forceps and carefully removed using a micro vessel cauterization tool. Mice were given 2.27 mg/ml Baytril in drinking water (CDMV, cat# 104674) as an antibiotic for 7 days post

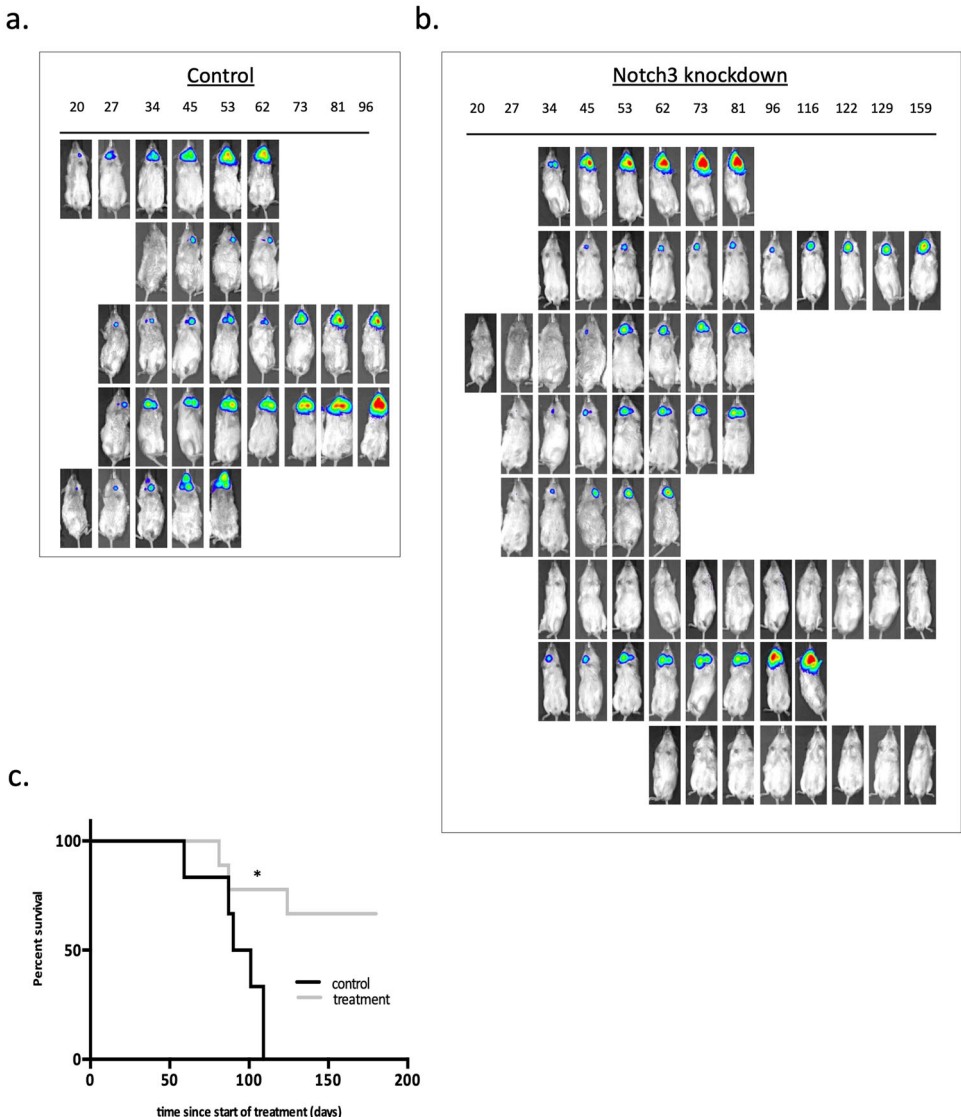

**Fig. 8 Inhibition of Notch signaling improves survival and inhibits growth of lymph node metastasis in an orthotopic mouse model of HNSCC.**
**a** UT-SCC-74A and UT-SCC-74B cells transduced with luciferase and with the inducible shRNA for Notch3 were injected submucosally into the tongues of the mice. After surgical removal of tongue tumors, images of lymph node metastasis of control (**a**) and doxycycline treated (**b**) mice were taken using the Xenogen IVIS-200 instrument. Numbers on top of the images represent number of days after the surgery. **c** The survival analysis of control and doxycycline treated mice are shown. Either poor health condition of the animal or luciferase signal of $1 \times 10^9$ at the site of metastasis was defined as an end point. Mantel-Cox test identified significant differences between the groups with improved survival of treated mice ($p$ value 0.0087).

surgery. Mice were weighed daily for 3 days post-surgery and weekly thereafter. Following surgery, mice were randomized into 2 groups and one group received 625 mg/kg doxycycline diet (Harlan). The establishment and growth of the metastatic lesions was monitored weekly using Xenogen IVIS-200 instrument.

**RNA extraction, retro transcription, and real-time PCR**. Total RNA was isolated from cells using the RNeasy Mini Kit (Qiagen Inc., Valencia, USA), following manufacturer's protocol. Retro transcription of isolated total RNA was carried out using qScript cDNA SuperMix (Quanta Biosciences, Gaithersburg, USA). Real-time PCR was carried out using the SYBR green PCR Mastermix (Applied Biosystems, Burlington, Canada). Quantification of RPL13 and HRPT1 transcripts were used to normalize template levels. All reactions were performed in triplicate for accuracy. Primers were designed for intron-spanning and/or overlap of exon-exon junction.

**Western blotting**. To obtain whole cell lysates, cells were washed with PBS and harvested in RIPA lysis buffer (G-Biosciences, St. Louis, USA) supplemented with 1× EDTA, phosphatase and protease inhibitors (Thermo Scientific). After incubation, samples were centrifuged at 4 °C for 30 min at 18,000CRF to remove cell debris. Protein concentration in the supernatant was measured using the Pierce bicinchoninic acid (BCA) assay kit (Thermo scientific). The whole cell lysates were

fractionated by SDS-PAGE and transferred to a nitrocellulose membrane using a Pierce G2 Fast Blotter transfer apparatus (Thermo Scientific). Membranes were incubated with 5% milk in PBST (0.1% Tween-20) for two hours to prevent non-specific binding of antibodies. Primary antibodies against β-actin (1:10,000) (MP Biomedicals, Santa Ana, California), Jag1, Jag2 (1:1000) (Cell Signaling Technology) were incubated overnight at 4 °C. Next, the membranes were washed three times for 5 min with PBST and incubated with 1:5000 diluted horseradish peroxidase conjugated (HRP) anti-mouse or anti-rabbit antibodies (GE Healthcare UK, Amersham, UK) for 2 h. Blots were washed three times with PBST and developed with Super-Signal West Pico Chemiluminescent Substrate (Thermo Scientific).

**CRISPR gene editing**. Genotyping for NM_000435.2: c.1939C>T was performed at The Centre for Applied Genomics, The Hospital for Sick Children, Toronto, Canada using a custom designed TaqMan® SNP Genotyping Assay (Life Technologies Corporation). The probe mixes consisted of PCR primers 5′-CAT-CAACCGCTACGACTGTGT 3′ (forward) and 5′-CCTCTCATGGCAGCCACTT 3′(reverse) and dual label TaqMan® MGB probes TGAAGCCAGGTTGGCAG (VIC) and TGAAGCCAGATTGGCAG (FAM). The 10 µl reaction mix consisted of 5 µl TaqMan Genotyping Master Mix (Life Technologies), 0.25 µl of 40X combined primer and probe mix, 2.75 µl water and 20–50 ng of DNA template. Cycling conditions for the reaction were 95 C for 10 min, followed by 40 cycles of

94 C for 15 s and 57 C for 1 min. Samples were analyzed using the ViiA™ 7 Real-Time PCR System (Life Technologies) and analyzed using ViiA™7 software.

**Histology**. TMA paraffin blocks were sectioned at 4 uM and stained at the histology lab of the UHN LMP-Pathology Research Program under the following conditions:

Hey1-Novus NBP2-47436, antigen retrieval with citrate pH6.0, 1/100 overnight incubation with the primary antibody

Hes1-Novus NBP2-67642, antigen retrieval with Tris-EDTA pH9.0, 1/3000 1 h incubation with the primary antibody

Jag1-Novus NBP2-66912, antigen retrieval with low temp citrate pH6.0, 1/200 1 h incubation with the primary antibody

Jag2-Novus NBP1-86337, antigen retrieval with citrate pH6.0, 1/200 1 h incubation with the primary antibody

Following primary antibody incubation, MACH 4 reagents (Intermedico Cat# BC-M4U534L) were applied as per kit instructions and the color was developed using DAB (DAKO Cat# K3468). Slides were counterstained lightly with Mayer's Hematoxylin, dehydrated in alcohols, cleared in xylene and mounted with MM 24 Leica mounting medium (Cat#3801120).

**Quantification and statistical analysis**. Statistical parameters are reported in the figures and figure legends. In figures, asterisks denote statistical significance as calculated by Student's $t$-test ($*p < 0.05$; $**p < 0.01$; $***p < 0.001$; $****p < 0.0001$). Statistical analysis for graph figures was performed in GraphPad PRISM 6.

**Reporting summary**. Further information on research design is available in the Nature Portfolio Reporting Summary linked to this article.

## Data availability

Data resources: Raw data files for the Illumina Gene Expression analysis have been deposited in the NCBI Gene Expression Omnibus under the accession number GSE117753. Uncropped blots are available as Fig. S7. Source data is available as Supplementary data 11.

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

## Acknowledgements

This research was supported by the Terry Fox frontiers program project grant.

## Author contributions

Conceptualization, M. Kondratyev and B.G.W; Investigation, M. Kondratyev, A.P., C.B, Z.S, S.M, M.B, P.B; Methodology, M. Kondratyev, T.K, J.M, C.B, A.Dvorkin-Gheva, M.B; Formal analysis, M. Kondratyev, A.S, T.K, N.S, C.V, S.S; Data curation, A.S, T.K, N.S, C.V, S.S, T.J.P, Writing - original draft, writing – review and editing, M. Kondratyev, S.M, M. Koritzinsky and B.G.W, Funding acquisition, M. Kondratyev and B.G.W; Resources, R.A.G, J.M, A.Datti; Supervision, M. Kondratyev, T.J.P, M. Koritzinsky and B.G.W.

## Competing interests

The authors declare no competing interests.
