## [Peer Review File · Communications Biology]

Reviewers' comments:

Reviewer #1 (Remarks to the Author):

The manuscript entitled "Identification of acquired Notch3 dependency in metastatic head and neck cancer" by Maria Kondratyev et al, described the analyses of three "matched" primary and metastatic pairs of HNSCC lines and examined the roles of differentially expressed Notch3 signaling molecules and mutated Notch3 in metastatic lines. Their findings have been validated in TMA from over 200 HNSCC patients, and showed that suppression of mutated Notch3 improved the mouse host survival bearing subcutaneously or orthotopically injected metastatic HNSCC. The investigation of Notch3 and related signaling molecules in metastatic HNSCC is interesting and important to understand the pathogenesis of tumor metastasis. However, the most findings in this study are descriptive, and deficiencies in the scientific presentation and manuscript writing make this manuscript unacceptable at the present form.

The specific comments are:

1. In Figure 1A, the authors described the primary locations of 3 paired HNSCC cell lines, and showed that primary tumors of patient #54, #60 and #74 arose from different anatomic sites (UT-SCC-54A - buccal mucosa, UT-SCC-60A - left tonsil and UT-SCC-74A- tongue). However, the graphic presentations of 54A and 60A tumors were pointed to the same location as in the left tonsil. Please check, explain, and make correction.
2. In the table presented in Fig 1A, the label of #54 primary tumor location was spelt as "buchal mucosa", which could be a typo. In this table, the source of the cell lines may not need to be included, because all lines were from the same lab and the source could be stated in the Method. The authors should include more important information in this table. Are those primary and metastatic cell lines were isolated at the same time, or from the different time points? Any prior treatment for patients was conducted before isolation of the cell lines? The patient's survival information?
3. In Figure 1A, the TMN staging and grade information are not explained clearly to readers who are not involved in HNSCC clinical study. Why the patient #54 had neck metastasis but with T2N0M0 stage, which indicates no lymph node and distance metastasis identified when diagnosed? Is the "Grade" referring to pathological grade regarding tumor cell differentiation status? Please explain in the figure legend. Any information of tumor infiltrating cells or inflammation status?
4. The origin of 60A/B cell lines was derived primarily from tonsil and exhibited very different molecular signature and functional behavior. These differences included the distinct top genetic defects showing in Fig 1B, decreased protein expression of Jag2 in 60B showing in Fig 2E, and failed modulation of Vimentin and E-Cadherin expression by Jag1 and Jag2 ligands in Fig 4C and D, etc. Please explain and discussion the potentially different mechanisms mediated by Notch signaling in this cell line when compared with other lines studied.
5. Most fig and table legends are not very informative, which create difficulties for readers to understand the points. For example, what is the meaning of the label of "PM" in fig 1B? All the Supplemental Tables have no table number and title labels on the excel file, except Suppl Table 1, so the readers do not know which excel file is which Suppl Table.
6. Fig 2B, among the top 17 upregulated pathways, 3 are involved with myeloid cell and/or leukocyte, and 3 are involved with adhesion and integrins. Please discussed the molecules involved in these pathways, and their regulatory mechanisms related to Notch pathways or their contribution to tumor metastasis studied in this manuscript. For Fig 2B and C, please remove GO numbers, which make the labels very long and confusion. Such GO numbers could be included in the supplemental table.
7. Fig 3A, it is not clear how many shRNA per gene was used for the screening? In Fig 3B, the gene name was labeled as PSEN1, however, in the text, it was called PSN1, please check and make it consistent. Fig 3C, it is not very clear what kind assay was presented? What does the

intensity mean? The calculation of intensity is from how many replicates? How the statistical significance was calculated?

8. Fig 3D is a very confusing graph. In the fig legend, the authors stated "D. Distributions of Notch3 essentiality scores across larger set of cell lines were analyzed using Expectation Maximization algorithm (60). All cell lines derived from the primary tumors belonged to the same population, while cell lines derived from metastases were divided into 2 sub-populations with low essentiality scores ("low subpopulation") and high essentiality scores ("high subpopulation")." First of all, it is not very clear what is the meaning of "larger set of line lines". How many cell lines the authors studied? Where is the information of these cell lines? Second, in the Methods, there is NO description of how the scores were calculated, and what is the meaning of "essential scores"? In this fig, what are those colored squares and lines between them meaning? Are those represented different cell lines? The colored legends DO NOT make any sense without detailed explanation. What are the two colored peaks meaning inside the box labeled "Frequency"? Please explain well about the "Notch3 essentiality scores", "Index" and "Frequency" presented in Fig 3D. Please explain clearly how you get these results and the meaning of the results using the language which average biologists are able to understand.

9. Fig 5A IHC of TMA is not well explained. In the Results, the authors stated "Using immunohistochemistry, we looked at the expression of Jag1, Jag2, Hes1 and Hey1 in the TMA. We determined that for each one of proteins the cores can be divided into "low" "intermediate" and "high" populations based on the percentage of positive cells within the sample; Jag1 only had a "high" and "low" populations (Figure 5A)." In the Fig legends, the authors stated "A. A distribution of frequencies of positive cells for each marker across the TMA cores was calculated based on the optimal number of Gaussians fitted by the EM algorithm as described in methods." What does the density of Y axis mean, and what is X axis of the first two bar graphs? What is the brown and green staining for? How the IHC images were scanned with what kind instruments and software? The images were collected from how many fields, or from the total area of the stained tissues? The IHC staining images were from how many magnification under the microscopy?

10. Fig 5B, please label the cased numbers of the different staining and disease status. Is any survival difference among these patients? For the legend of "Figure 5. Differential expression of Notch pathway components in TMA constructed from over 400 HNSCC patient tumors". However, in the abstract, the authors stated "These components were also shown to be differentially expressed between early and late stages of tumors in a TMA constructed from over 200 HNSCC patients." The TMA had 200 or 400 patient's tumors? Please check, explain, and be consistent.

11. It seems that both wild type and mutated Notch3 identified in this study promoted tumor growth. Is any functional difference between wt and mt Notch3? Does the mutation make Notch3 more oncogenic? Please discuss.

12. The authors claimed that they identified a novel mutation on Notch3 in the EGF-like repeat 16 domain. However, there is no mechanistic study to explain how this mutation affected Notch3 tumor promoting activity. Has this Notch3 mutation been identified from HNSCC TCGA datasets? Since TCGA datasets have HNSCC patient's lymph node information, the author could analyze if this Notch3 mutation is correlated with HNSCC lymph node status and affect patient's survival.

Reviewer #2 (Remarks to the Author):

In the submitted manuscript, Kondratyey et al investigated components of NOTCH3 signaling in matched pairs of HNSCC cell lines derived from primary tumors and metastatic sites, tumor samples from HNSCC patients and xenograft mouse models. Though the principle concept of this study is of high clinical relevance addressing an unmet medical need in the field of interest, several conclusions are not supported by sufficient amount of experimental data and additional experiments are required to improve the quality and impact of the manuscript.

Major points

1. Figure 1 summarizes the mutational profiling of HNSCC cell lines derived from primary tumors

and lymph node metastasis. However, the number of matched pairs (n=3) is rather low and provided data require confirmation by analysis of independent and newly established primary cultures of matched pairs, to avoid any bias by long-term in vitro culture.

2. Fig. 2B-C include several GO terms, which are not statistically significant ($p > 0.05$), including the NOTCH signaling pathway (see Table S2).

3. Fig. 2F demonstrates differences in Vimentin and E-cadherin transcript levels indicating EMT in cell lines derived from metastatic sites. Data should be confirmed on protein level and their expression assessed in cell lines with silenced NOTCH3 or JAG2 expression as well as tumor samples from mouse models and on TMAs. Did the authors observe any change in EMT-related cellular (e.g. cell morphology, motility or invasion) or molecular traits (e.g. expression of EMT-related transcription factors) upon silencing of NOTCH3 or JAG2? This is of particular importance as a recent study reported repression of NOTCH3 by ZEB1 to permit NOTCH1-mediated EMT in SCC (PMID: 29170450).

4. A $FDR < 0.2$ in Fig. 3B cause a substantial risk for false-positive candidates and I wonder, how many components of the NOTCH pathway remain statistically significant applying a $FDR < 0.05$?

5. How did the authors confirm the silencing efficacy in Fig. 3E-F, and had silencing of JAG2 or NOTCH3 any impact on expression of other NOTCH proteins? Those data should be presented as Supplemental Figure and are relevant, as a recent study reported induction of EMT in head and neck cancer by NOTCH4-HEY1 signaling (PMID: 29146722).

6. Fig. 4C-D demonstrate altered Vimentin and E-cadherin expression upon JAG1 or JAG2 stimulation in vitro. However, JAG1-induced down-regulation of Vimentin accompanied by up-regulation of E-cadherin was only statistically significant for one out of three cell lines derived from primary tumors. Moreover, authors should repeat the experiment under conditions of constitutive or even better conditional NOTCH3 silencing.

7. Fig. 5 demonstrates IHC staining of downstream NOTCH components on a TMA and their association with clinical data. What was the rationale not to include IHC staining for NOTCH proteins, in particular NOTCH3? Authors should provide data of a crosstab analysis for all available demographic, clinical and histopathological data, including pathological grading and lymph node metastasis as Supplemental Table. Had the expression of NOTCH components any prognostic value considering overall, disease-specific or progression-free survival in univariate and multivariate COX regression analysis?

8. Is the mutation shown in Fig. 1E and analyzed in more detail in Fig. 6 listed in COSMIC or any other somatic mutation database?

9. How did the authors confirm conditional silencing efficacy of NOTCH3 in xenograft models shown in Fig. 7-8? Did tumors or samples from lymph node metastasis with Dox treatment exhibit reduced expression of JAG2 or other components of NOTCH signaling and affect expression of EMT-related genes?

Minor Points

1. Previous studies reporting differences in the genomic landscape or global gene expression profiles between HNSCC patients with or without lymph node metastasis or matched pairs of primary tumors and lymph node metastasis did not demonstrate a critical impact for components of NOTCH3 signaling, which should be mentioned and critically discussed by the authors.

Reviewer #1 (Remarks to the Author):

The manuscript entitled “Identification of acquired Notch3 dependency in metastatic head and neck cancer” by Maria Kondratyev et al, described the analyses of three “matched” primary and metastatic pairs of HNSCC lines and examined the roles of differentially expressed Notch3 signaling molecules and mutated Notch3 in metastatic lines. Their findings have been validated in TMA from over 200 HNSCC patients and showed that suppression of mutated Notch3 improved the mouse host survival bearing subcutaneously or orthotopically injected metastatic HNSCC. The investigation of Notch3 and related signaling molecules in metastatic HNSCC is interesting and important to understand the pathogenesis of tumor metastasis. However, the most findings in this study are descriptive, and deficiencies in the scientific presentation and manuscript writing make this manuscript unacceptable at the present form.

The specific comments are:

1. In Figure 1A, the authors described the primary locations of 3 paired HNSCC cell lines, and showed that primary tumors of patients #54, #60 and #74 arose from different anatomic sites (UT-SCC-54A - buccal mucosa, UT-SCC-60A - left tonsil and UT-SCC-74A- tongue). However, the graphic presentations of 54A and 60A tumors were pointed to the same location as in the left tonsil. Please check, explain, and make correction.

The graphic presentation contained an error that have been corrected.

2. In the table presented in Fig 1A, the label of #54 primary tumor location was spelt as “buchal mucosa”, which could be a typo. In this table, the source of the cell lines may not need to be included, because all lines were from the same lab and the source could be stated in the Method. The authors should include more important information in this table. Are those primary and metastatic cell lines were isolated at the same time, or from the different time points? Any prior treatment for patients was conducted before isolation of the cell lines? The patient’s survival information?

We took the table out of the figure and included references to the cell line origins instead. In general, the time of isolation varies between the cell lines. For some patients, the cell lines come from the same surgery, for others they don’t. For example, for patient 74, the UT-SCC-74A cell line was established from the primary tumor. After a rather short interval the patient has difficulties with swallowing and speech. On the contralateral left side, a suspicion of a neck metastasis was based on a CT examination. A tracheostomy, hypopharyngoscopy and neck dissection to the left was performed. The UT-SCC-74B cell line was established from this biopsy. All the questions regarding the origin of cell lines as well as survival information for the patients can be directed to Dr. Grenman at reigre@utu.fi.

3. In Figure 1A, the TMN staging, and grade information are not explained clearly to readers who are not involved in HNSCC clinical study. Why the patient #54 had neck metastasis but with T2N0M0 stage, which indicates no lymph node and distance metastasis identified when diagnosed? Is the “Grade” referring to pathological grade regarding tumor cell differentiation status? Please explain in the figure legend. Any information of tumor infiltrating cells or inflammation status?

As suggested, we are not including the table in the figure anymore thus, figure legend is not being changed. Instead, we are referencing the source of the lines. Patient 54 did not have neck metastasis at time of the diagnosis thus the T2N0M0 stage when the primary tumor was found. While not in the scope of this publication, it might be worth mentioning at this point, that patient 54 had a recurrence following

resection of primary tumor at which time neck metastasis was also found. Thus, in addition to 54A cell line we also have 54B cell line developed from the recurrence and 54C cell line developed from the neck metastasis. The grade refers to pathological grade indicating the differentiation and the proliferation status of the tumor.

4. The origin of 60A/B cell lines was derived primarily from tonsil and exhibited very different molecular signature and functional behavior. These differences included the distinct top genetic defects showing in Fig 1B, decreased protein expression of Jag2 in 60B showing in Fig 2E, and failed modulation of Vimentin and E-Cadherin expression by Jag1 and Jag2 ligands in Fig 4C and D, etc. Please explain and discuss the potentially different mechanisms mediated by Notch signaling in this cell line when compared with other lines studied.

We do believe that it is an interesting objective to explore differential phenotypes of HNSCC cells from different sites. We are currently working with a collection of HNSCC cells derived from multiple patients that include tumors from various origins (tongue, floor of the mouth, nostril and more). In a future publication we plan to discuss differences and similarities between functional dependencies, drug response and gene expression of various sites of HNSCC tumors.

60B cells do exhibit different response in several assays compared to 54C and 74B cell lines. However, similar to other metastatic cells used in this study, they exhibit functional dependency on Notch3, and Jagged 2, mesenchymal phenotype based on Vimentin/E-cadherin expression and downregulation of Jagged1 and Hes1 accompanied by high levels of Hey1. Moreover, while perhaps to the lesser extent than 54s and 74s, 60 cells are also susceptible to changes in expression of Notch target genes when cultured in the presence of Notch ligands. All these observations provide a compelling rationale to include data obtained with 60B cells in the scope of this publication.

5. Most fig and table legends are not very informative, which create difficulties for readers to understand the points. For example, what is the meaning of the label of “PM” in fig 1B? All the Supplemental Tables have no table number and title labels on the excel file, except Suppl Table 1, so the readers do not know which excel file is which Suppl Table.

We have changed the labeling in figure 1B and it now stated “primary” instead of “P” and “metastatic” instead of “M”. We also added table numbers within the excel files for supplementary tables.

6. Fig 2B, among the top 17 upregulated pathways, 3 are involved with myeloid cell and/or leukocyte, and 3 are involved with adhesion and integrins. Please discuss the molecules involved in these pathways, and their regulatory mechanisms related to Notch pathways or their contribution to tumor metastasis studied in this manuscript. For Fig 2B and C, please remove GO numbers, which make the labels very long and confusion. Such GO numbers could be included in the supplemental table.

We included discussion of the possible correlation of myeloid cell/leukocyte as well as adhesion/integrins pathways with the Notch signalling (lines 165-178).

“It has been demonstrated by many groups that Notch pathway plays a key role in differentiation which might be correlated with its role in tumorigenesis. For example, inhibition of Notch signaling in hematopoietic progenitor cells (HPC), myeloid-derived suppressor cells (MDSC), and dendritic cells is directly involved in abnormal myeloid cell differentiation in cancer (42). Activation of Notch signaling in

macrophages led to secretion of CCL5, one of the genes found to be upregulated in our metastatic cell collection and led to increased EMT and tumor cell migration (43).

Growing evidence supports a role of Notch interactions with integrins in cancer progression. For example, it has been demonstrated that Notch signaling activates integrin b1 thus enhancing cell adhesion of cancer cells entering blood circulation (44). Moreover, expression of Jagged-1 was shown to be dependent on several integrin subunits (45). Overall, the evidence suggests that the cross talk between Notch and integrins is context dependent and is largely regulated by the microenvironment. Thus, it plausible that a cross talk between Notch3/Jag2 axis and integrins plays a role in HNSCC metastasis which will be explored in further publications. “

We removed GO numbers from figures 2B and 2C and included them in supplementary table S2.

7. Fig 3A, it is not clear how many shRNA per gene was used for the screening? In Fig 3B, the gene name was labeled as PSEN1, however, in the text, it was called PSN1, please check and make it consistent. Fig 3C, it is not very clear what kind assay was presented? What does the intensity mean? The calculation of intensity is from how many replicates? How was the statistical significance calculated?

Each gene was targeted by average of 5 different shRNAs. We included this information in the method section (lines 901-931).

“Near whole-genome shRNA dropout screens were carried out using the lentiviral 80K shRNA library (62). Each gene was targeted by average of 5 different shRNAs. Cell populations were analyzed after ~3 and ~6 population doublings following infection and the siMEM hierarchical regression algorithm which models the level of each shRNA as a regression function of the initial abundance of this shRNA (population at time T0), baseline trend in this shRNA’s abundance across time points and difference in abundance trend between samples sharing a common feature. In short, this model represents the time course as a line with an intercept and a slope (62); the slope reflecting the rate of the dropout of the shRNA and termed “essentiality score”. The siMEM algorithm was also applied to test for differences in gene essentiality between matched pairs of cell lines derived from primary and metastatic sites which yielded p-values indicating statistical significance of these differences. “

We corrected the name of the gene in figure 3B to PSN1.

We changed the legend for figure 3C to better explain the assay and the calculations (lines 728-733) and we added more detailed explanation on how the statistical significance is calculated (see above).

“C. The abundance of NOTCH3 shRNA is reflected by the fluorescence intensity in the Affymetrix array measured across the 3 time points of the screen (0, 1 and 2, corresponding to T0, T1 and T2, respectively). For each patient there are 2 plots, first showing the result from the primary tumor sample and the second showing the result from the metastatic sample from the same patient. Each time point in each of the plots is represented by 3 replicates.”

8. Fig 3D is a very confusing graph. In the fig legend, the authors stated “D. Distributions of Notch3 essentiality scores across larger set of cell lines were analyzed using Expectation Maximization algorithm (60). All cell lines derived from the primary tumors belonged to the same population, while cell lines derived from metastases were divided into 2 sub-populations with low essentiality scores (“low subpopulation”) and high essentiality scores (“high subpopulation”).” First of all, it is not very clear what is the meaning of “larger set of line lines”. How many cell lines the authors studied? Where is the

information of these cell lines? Second, in the Methods, there is NO description of how the scores were calculated, and what is the meaning of “essential scores”? In this fig, what are those colored squares and lines between them meaning? Are those represented different cell lines? The colored legends DO NOT make any sense without detailed explanation.

What are the two colored peaks meaning inside the box labeled “Frequency”? Please explain well about the “Notch3 essentiality scores”, “Index” and “Frequency” presented in Fig 3D. Please explain clearly how you get these results and the meaning of the results using the language which average biologists are able to understand.

We included supplementary table 4 that has information about all the cell lines used in figure 3D. We also included the information about the number of cell lines and explained the colored squares and lines in details in the legend for figure 3D (lines 733-746) . The term index means “cell lines” that are plotted in order across the plot. The terms index and frequency are also now explained in the legend; the term “essentiality score” is now explained in the method section describing the analysis of shRNA screens.

“Notch3 essentiality scores across a set of 29 cell lines (see Table S5 for details) are shown. The x axis (labeled index) represents the ordered list of the cell lines shown in the plot. Samples are paired based on the patient they were obtained from, and each pair is connected by a gray line. Cell lines derived from the primary tumors are contoured with a blue line; cell lines derived from the metastatic samples are contoured with red lines. “Pri” – primary tumor derived cell lines ; “met” – metastases derived cell lines; “HPV+” HPV positive cell lines ; “rec” – recurrence derived cell lines. Insert: Distributions of Notch3 essentiality scores of primary tumors derived cell lines (blue) and metastases derived cell lines (red) are shown on a density plot that shows frequency of each score in each cell line.

For cell lines derived from metastatic samples, the distribution of essentiality scores is bimodal, allowing to separate the cell lines into 2 groups (see red contours). Comparison between these 2 groups showed a significant difference (Welch Two Sample t-test, $p = 4.029e-05$); additionally, the bottom group of metastatic samples was significantly different from the primary tumors (Welch Two Sample t-test, $p = 1.555e-06$).”

9. Fig 5A IHC of TMA is not well explained. In the Results, the authors stated “Using immunohistochemistry, we looked at the expression of Jag1, Jag2, Hes1 and Hey1 in the TMA. We determined that for each one of proteins the cores can be divided into “low” “intermediate” and “high” populations based on the percentage of positive cells within the sample; Jag1 only had a “high” and “low” populations (Figure 5A).” In the Fig legends, the authors stated “A. A distribution of frequencies of positive cells for each marker across the TMA cores was calculated based on the optimal number of Gaussians fitted by the EM algorithm as described in methods.” What does the density of Y axis mean, and what is X axis of the first two bar graphs? What is the brown and green staining for? How the IHC images were scanned with what kind instruments and software? The images were collected from how many fields, or from the total area of the stained tissues? The IHC staining images were from how many magnification under the microscopy?

We included details of IHC staining in the method section (lines 930-938).

“Each TMA was imaged at 40x magnification using an Aperio ScanScope AT2. The digitized images, each containing approximately 200 individual cores, were loaded into Visiopharm software (version 2019.10, Visiopharm, Hoersholm, Denmark), in which each core was isolated and analyzed separately. The nuclear counterstain was used to identify a cell nucleus, and a cytoplasm was simulated. To visualize the segmented cores, an overlay showing nuclear segmentation (light blue), negative cytoplasm (green) and positive

cytoplasm (red) is shown in figure 5 below a screenshot of the original core image. Statistical outputs relating to the number of cells positive for particular markers, and the brown intensity within those cells, were exported for subsequent statistical analysis. “

We changed the legend for figure 5A to include explanation for the y and x axis titles (lines 764-772).

“4 uM sections of the TMA were stained with antibodies against the components of the Notch pathway previously identified as differentially expressed by metastatic HNSCC lines. A. Distributions of fractions of positive cells for Jag1, Jag2, Hes1 and Hey1 across the TMA cores. Y-axis on the left axis indicates Density values, related to the density curve (the density curve is showing the smoothed distribution of the points along the x-axis). Y-axis on the right indicates frequency values related to the histogram. Thresholds between the subpopulations obtained by using EM algorithm (see Methods) are indicated with vertical black lines; the exact values of the thresholds are indicated near the corresponding lines.”

10. Fig 5B, please label the cased numbers of the different staining and disease status. Is any survival difference among these patients? For the legend of “Figure 5. Differential expression of Notch pathway components in TMA constructed from over 400 HNSCC patient tumors”. However, in the abstract, the authors stated “These components were also shown to be differentially expressed between early and late stages of tumors in a TMA constructed from over 200 HNSCC patients.” The TMA had 200 or 400 patient’s tumors? Please check, explain, and be consistent.

The cased numbers along with their clinical information including survival are now summarized in supplementary table S6. We clarified the confusion – the TMA contains about 400 tumors however we needed to perform several filtering steps that left us with about 200 cores. The explanation for the steps is now included in result section (lines 303-305) and in figure S2.

“Cores that had obvious artifacts due to edge effects, air bubbles, etc. were eliminated from the analysis. Regions that contained non-tumor tissue such as normal salivary gland or muscle were also excluded (Figure S2).”

11. It seems that both wild type and mutated Notch3 identified in this study promoted tumor growth. Is any functional difference between wt and mt Notch3? Does the mutation make Notch3 more oncogenic? Please discuss.

According to our hypothesis, a novel mutation identified in Notch3 in 74B contributes to metastatic phenotype of the cells. This hypothesis is supported by increased expression of Vimentin and decreased expression of E-Cadherin in 74B cells compared to 74A and the fact that we were able to convert this phenotype to the one of 74A by correcting the mutation. Additionally, we think that the mutation contributed to the dependency of the cells to Notch3 Jag2 signaling, a feature that we find as unique in metastatic cells. As to the rate of growth, we do not see significant differences in growth rate and take of 74A and 74B cells in subcutaneous mouse model. In vitro, 74B has a higher rate of proliferation than 74A and correcting the mutation does result in slowing the rate and making cells “more similar” to 74A. We do not have a compelling reason to conclude that 74B is more tumorigenic than 74A; those are both cell lines derived from actively growing tumors. The big difference is that 74B went through adaptations that allowed these cells to survive and proliferate at the metastatic site, which, based on our data include mutation in Notch3, elevated Jag2 and Hey1 expression and induction of EMT.

12. The authors claimed that they identified a novel mutation on Notch3 in the EGF-like repeat 16

domain. However, there is no mechanistic study to explain how this mutation affected Notch3 tumor promoting activity. Has this Notch3 mutation been identified from HNSCC TCGA datasets? Since TCGA datasets have HNSCC patient's lymph node information, the author could analyze if this Notch3 mutation is correlated with HNSCC lymph node status and affect patient's survival.

This is a novel mutation not previously reported which is not surprising because all the TCGA data sets contain data from primary tumors. Even with the lymph node status known, the sequencing itself was not performed on the metastatic tissue, thus we do not expect this mutation to be present. We hypothesize that the described mutation appeared during metastatic transformation and contributes to the ability of metastatic cells to survive and growth at the metastatic site. Figure 6 demonstrates that correcting the mutation results in reversion of the metastatic phenotype of 74B cells, confirming that the mutation was indeed contributing to metastatic transformation of the primary tumor in patient # 74. Several other mutations in EGF domains of Notch's were previously reported as discussed in the discussion section of the manuscript. Those might have similar mechanistic consequences as the mutation identified here.

Reviewer #2 (Remarks to the Author):

In the submitted manuscript, Kondratyey et al investigated components of NOTCH3 signaling in matched pairs of HNSCC cell lines derived from primary tumors and metastatic sites, tumor samples from HNSCC patients and xenograft mouse models. Though the principle concept of this study is of high clinical relevance addressing an unmet medical need in the field of interest, several conclusions are not supported by sufficient amount of experimental data and additional experiments are required to improve the quality and impact of the manuscript.

Major points

1. Figure 1 summarizes the mutational profiling of HNSCC cell lines derived from primary tumors and lymph node metastasis. However, the number of matched pairs (n=3) is rather low and provided data require confirmation by analysis of independent and newly established primary cultures of matched pairs, to avoid any bias by long-term in vitro culture.

In addition to the whole exome sequencing analysis performed on the 6 cell lines described above, we captured and sequenced the most commonly altered genes in HNSCC as reported by the TCGA in cell lines derived from multiple HNSCC patients (Figure 1F, Table S2, Table S3). While a comprehensive analysis of this work will be presented in a future publication, here we report the data from 3 additional patients that had matched lines derived from primary tumors and metastasis. Comparing mutational profiles of these patients between the primary tumor derived cells and their metastatic counterparts, we observed alterations on several genes related to Notch signaling that are present exclusively in metastases. These observations further suggest that genetic alterations in Notch signaling components play a role in the metastatic transformation of HNSCC cells (lines 141-151).

In addition to the whole exome sequencing analysis performed on the 6 cell lines described above, we captured and sequenced the most commonly altered genes in HNSCC as reported by the TCGA in cell lines derived from multiple HNSCC patients (Figure 1F, Table S2, Table S3). While a comprehensive analysis of this work will be presented in a future publication, here we report the data from 3 additional patients that had matched lines derived from primary tumors and metastasis. Comparing mutational profiles of these patients between the primary tumor derived cells and their metastatic counterparts, we observed alterations on several genes related to Notch signaling that are present exclusively in metastases. These include Notch receptors Notch1 and Notch4 as well as an EP300 gene that was shown to be associated with

Notch signaling pathway (41). These observations further suggest that genetic alterations in Notch signaling components play a role in the metastatic transformation of HNSCC cells.

2. Fig. 2B-C include several GO terms, which are not statistically significant ($p > 0.05$), including the NOTCH signaling pathway (see Table S2).

Our team of bioinformaticians provided with the following response:

The choice of an appropriate significance level during statistical testing has always been somewhat subjective in nature, with the most common choices being .01, .05 and .1. The concept of statistical significance was introduced by RA Fisher in the 1930's and became standardized predominantly through the use of standardized calculation tables that were provided to researchers who did not necessarily wish to calculate a p-value directly. The p-value (or modified Fisher test EASE score in this case) is of course dependent on the spread of the data, the size of the sample, and the effect size and in the case of a Fisher test it is an "exact test" in that it gives an exact p-value as a result. The controversy over the use, and misuse, of p-values has been so large in recent years that in 2016 the American Statistical Association put out a statement on p-values. The most salient point in their principles section related to the comments received by the reviewers would be that "P-values do not measure the probability that the studied hypothesis is true, or the probability that the data were produced by random chance alone." It is important to contextualize the EASE score and p-value together with the experimental design and other measurements that were performed in our study to draw out an overall conclusion as to the validity of the study. In the current case, we believe that this picture shows an overall view of Notch related signaling that is reflected in an enrichment of genes in some key pathways. It should be noted that while the authors note that the EASE score in their example at .06 is considered "not significant", this is again highly subjective and in fact, the default setting for users is an EASE score of .1. We choose this to mean that scores $< .1$ should be viewed with caution (not necessarily discarded) and need to be backed up by other parts of the study to paint the full picture. To be certain, a p-value (or EASE score) of $p < 0.1$ is not unheard of within the gene expression literature, and again, it is more important to have the exact value presented and let the reader determine whether the enrichment of the pathways seen here are congruent with the overall findings and conclusions of the paper. Were the p-values or EASE scores the only evidence presented, we would surely agree with the reviewers' statements, and indeed, would likely feel that the overall hypothesis presented here would not pass muster even if the score values were less than .01 if none of the other evidence presented validated this overall picture.

Wasserstein RL, Lazar NA. The ASA's statement on p-values: context, process, and purpose, The American Statistician. 2016; DOI:10.1080/00031305.2016.1154108 *Buying a significant result: Do we need to reconsider the role of the P value?*

Michael R. Jiroutek DrPH, MS J. Rick Turner PhD, DSc

First published: 26 May 2017 <https://doi.org/10.1111/jch.13021> Bangdiwala SI. Understanding Significance and P-Values. Nepal J Epidemiol. 2016;6(1):522-524. Published 2016 Mar 31.

doi:10.3126/nje.v6i1.14732 *Dahiru T. P - value, a true test of statistical significance? A cautionary note. Ann Ib Postgrad Med. 2008;6(1):21-26. doi:10.4314/aipm.v6i1.64038*

3. Fig. 2F demonstrates differences in Vimentin and E-cadherin transcript levels indicating EMT in cell lines derived from metastatic sites. Data should be confirmed on protein level and their expression assessed in cell lines with silenced NOTCH3 or JAG2 expression as well as tumor samples from mouse models and on TMAs. Did the authors observe any change in EMT-related cellular (e.g. cell morphology, motility or invasion) or molecular traits (e.g. expression of EMT-related transcription factors) upon silencing of

NOTCH3 or JAG2? This is of particular importance as a recent study reported repression of NOTCH3 by ZEB1 to permit NOTCH1-mediated EMT in SCC (PMID: 29170450.)

A modified figure 2F now includes data of protein expression of Vimentin and E-cadherin confirming the qPCR data (lines 203-208).

“EMT is associated with change in expression of several genes encoding cytoskeletal components of the cells; high expression of vimentin is normally associated with mesenchymal phenotype while E-cadherin is a known marker of epithelial cells. In line with this hypothesis, we observed higher expression of E-cadherin in cell lines derived from primary tumors while vimentin was highly expressed in metastasis-derived cell lines (Figure 2F).”

Expression of the genes was also measured in subcutaneous tumors seeded by 74A and 74B cells, with and without knockdown of Notch3 (figure S6)(lines 356-358).

Confirmation of knockdown efficiency and expression of Notch pathway components and EMT markers in tumors are shown in Figure S6A,B.

We did not have enough TMA material to stain for the EMT markers and we believe it would be difficult to make a reliable conclusion as the TMA does not contain metastatic tissue and there are a lot of factors contributing to expression of Vimentin and Cadherin in human tissues.

Cells derived from metastatic sites have dependency on Notch3 and Jag2 thus is it impossible to observe their phenotype/gene expression upon silencing – most the cells are dead.

4. An FDR<0.2 in Fig. 3B cause a substantial risk for false-positive candidates and I wonder, how many components of the NOTCH pathway remain statistically significant applying a FDR<0.05?

We modified the graph and now it shows genes with the FDR <0.05. All Notch pathway components remained statistically significant.

5. How did the authors confirm the silencing efficacy in Fig. 3E-F, and had silencing of JAG2 or NOTCH3 any impact on expression of other NOTCH proteins? Those data should be presented as Supplemental Figure and are relevant, as a recent study reported induction of EMT in head and neck cancer by NOTCH4-HEY1 signaling (PMID: 29146722.)

We confirmed the silencing efficacy by qPCR and we included this data now in figure S1.

As discussed in point 3, it is not possible to obtain good quality expression data from the metastatic cells upon silencing of Notch3 as they are highly dependent on expression of this gene.

6. Fig. 4C-D demonstrate altered Vimentin and E-cadherin expression upon JAG1 or JAG2 stimulation in vitro. However, JAG1-induced down-regulation of Vimentin accompanied by up-regulation of E-cadherin was only statistically significant for one out of three cell lines derived from primary tumors. Moreover, authors should repeat the experiment under conditions of constitutive or even better conditional NOTCH3 silencing.

As aforementioned, we could not do much with metastatic cells when Notch3 is silenced as they are dying very quickly. However, we do have multiple effects of Jag ligands on EMT, including downregulation of Vimentin in 2 out of 3 metastatic lines cultured on Jag1, and upregulation of Vimentin in 2 out of 3 primary

tumor derived lines cultured on Jag2. We do not in fact anticipate a large effect of Jag1 on primary tumor derived cell lines as they already express high levels of this ligands.

7. Fig. 5 demonstrates IHC staining of downstream NOTCH components on a TMA and their association with clinical data. What was the rationale not to include IHC staining for NOTCH proteins, in particular NOTCH3? Authors should provide data of a crosstab analysis for all available demographic, clinical and histopathological data, including pathological grading and lymph node metastasis as Supplemental Table. Had the expression of NOTCH components any prognostic value considering overall, disease-specific, or progression-free survival in univariate and multivariate COX regression analysis?

We do not see differences in expression of Notch proteins including Notch3 between primary and metastatic cells – we only see differences in its essentiality. The proteins that were differentially expressed in the cell lines are Jag1, Jag2, Hes1, Hey1 thus we stained the TMA for these markers. However, to address this comment we stained the TMA for Notch3, and the data is now included in figure S3. We also now included table S8 with the crosstab analysis and figure S3/ table S9 with the overall and disease-free survival analysis, as suggested.

8. Is the mutation shown in Fig. 1E and analyzed in more detail in Fig. 6 listed in COSMIC or any other somatic mutation database?

The mutation is not reported in the databases which is not surprising as we believe that this mutation only appears in metastatic tissues and the available mutations databases are mostly derived from the primary tumors. We did see however, 2 other mutations in Notch3 reported in cells derived from metastatic samples as discussed in the introduction (lines 67-71) (1).

“While most mutational and expression profiling data in HNSCC has been performed on primary tumors, recent analysis of metastasis derived samples suggested increased alterations in Notch signaling and reported mutations in Notch3 gene (7). These observations suggest a dual role of Notch in HNSCC, which is context dependent and needs to be further investigated. “

9. How did the authors confirm conditional silencing efficacy of NOTCH3 in xenograft models shown in Fig. 7-8? Did tumors or samples from lymph node metastasis with Dox treatment exhibit reduced expression of JAG2 or other components of NOTCH signaling and affect expression of EMT-related genes?

Figure S5-C confirms conditional silencing efficiency of Notch3 in the orthotopic model. We did not have enough material from the lymph nodes to perform qPCR on additional markers from the orthotopic model. However, Figures S5A and S5B now contain qPCR data confirming conditional silencing efficacy of Notch3 in subcutaneous xenografts as well as expression of Notch target genes and the EMT markers in those tumors.

Minor Points

1. Previous studies reporting differences in the genomic landscape or global gene expression profiles between HNSCC patients with or without lymph node metastasis or matched pairs of primary tumors and lymph node metastasis did not demonstrate a critical impact for components of NOTCH3 signaling, which should be mentioned and critically discussed by the authors.

We elaborated on it in the introduction (lines 67-71).

“While most mutational and expression profiling data in HNSCC has been performed on primary tumors, recent analysis of metastasis derived samples suggested increased alterations in Notch signaling and reported mutations in Notch3 gene (7). These observations suggest a dual role of Notch in HNSCC, which is context dependent and needs to be further investigated. “

A.

B.

F.

C.

D.

E.

Figure 1. Mutational profiling of matched HNSCC lines reveals a subset of mutations acquired upon metastasis.

Panel A is corrected with arrows pointing to the right sites of tumors. Table that contained clinical information of the patients is excluded.

Panel B is re-labeled stating Met and primary instead of M and P. Panel F is added with the results of targeted sequencing

Figure 2. Differential expression of Notch pathway components is observed in metastatic HNSCC lines. In B, C GO numbers were taken out of the graphs. Panel F was added with Western Blot analysis of Vimentin and E-Cadherin expression.

Figure 3. Pooled screens reveal metastasis-specific vulnerabilities. Panel D now shows distributions with $FDR < 0.05$ Panel D is modified according to suggestions of reviewers – see legend.

A

B

Figure S1. High throughput fluorescent imaging was utilized to evaluate the effects of knocking down of Notch3 on the growth of HNSCC cells derived from primary tumors and metastases. Panel B is new – silencing of the genes was confirmed by qPCR.

Figure S2. Characterization and filtering of the TMA. A new figure explaining filtering of samples in the TMA.

Figure S3. Notch3 expression in the TMA. A new figure showing expression of Notch3 in the TMA as suggested by reviewers.

Figure S4. Overall and disease-free survival analysis based on expression of Notch pathway components. A new figure asked by reviewers.

Figure S6. Growth quantification of lymph node metastasis and Notch knockdown confirmation in mice. Panels A and B are included as suggested by reviewers to show expression of genes in mouse tumors.

Reviewers' comments:

Reviewer #2 (Remarks to the Author):

Authors addressed most questions/concerns to improve the quality and impact of the manuscript. However, some issues are not adequately or insufficiently addressed and should be clarified prior publication:

1. Please harmonize presentation style for data in Fig. 1B and 1F or even better integrate them in one graph.

2. Despite the detailed response provided by the bioinformaticians, it remains a critical issue that the NOTCH signaling pathway is not ranked within the top 10 GO terms (Fig. 2B) and did not reach statistical significance. Authors should mention these facts in the main text and discuss them as potential limitations.

3. Revised Fig. 2F is confusing and detailed information is missing. What is the difference between the upper and lower graphs or Western blots? Western blots are incomplete: vimentin is missing for the upper and eIF4e for the lower. Moreover, protein expression patterns appear inconsistent with transcript levels. Though not explained at the figure legend * most likely indicates statistical significance ($p < 0.05$), which is questionable for 60 (lower left, vimentin) and 74 (lower right, E-cadherin), considering presented mean values + SEM. How many independent biological replicates were used to compute mean values + SEM?

Figure S6A-B shows Vimentin and E-cadherin transcript levels (most likely label of the y-axis last graph in B is wrong) assessed by RQ-PCR. Presented data do not support in vitro findings and are biased by potential differences in the relative composition of the tumor by cancer and stromal cells. Hence, RQ-PCR cannot replace IHC or co-IF staining with tumor sections, and the conclusion on EMT and its reversal by NOTCH3 and its ligands in primary tumor and metastasis-derived cell lines are not supported by sufficient experimental evidence.

4. Please modify the statement at page 225-229 as DLL3 and PSEN1 did not reach $FDR < 0.05$ and other candidates (NOTCH3, HEYL, LCK) were not within the top 30 differentially essential genes concerning the relative dropout and not within the top 200 concerning FDR.

5. Authors stress several times in the rebuttal that metastasis-derived cells die upon NOTCH3 (or JAG2) silencing, which hamper further analysis (e.g. EMT-related morphology/gene expression or expression of other NOTCH receptors upon NOTCH3 or JAG2 silencing): "Cells derived from metastatic sites have dependency on Notch3 and Jag2 thus is it impossible to observe their phenotype/gene expression upon silencing – most the cells are dead", "As discussed in point 3, it is not possible to obtain good quality expression data from the metastatic cells upon silencing of Notch3 as they are highly dependent on expression of this gene" and "we could not do much with metastatic cells when Notch3 is silenced as they are dying very quickly". However, data presented in Fig. 3E and 3F demonstrate reduced growth rates over time, but not such a severe phenotype, and obviously good quality expression data were obtained to demonstrate silencing efficacy in Figure S1. Please clarify/explain conflicting statements.

6. Figure S4 and Table S9 do not support any clinical relevance of downstream NOTCH components in the analyzed cohort, and Table S8 is inadequate to address my question. The crosstab should present association between relevant variables (e.g. pathological grading or lymph node metastasis) with high or low protein expression of JAG1, JAG2, HES1 or HEY1 to support experimental findings from cell culture and mouse models in patient tumor specimens. Table S9 is cited at line 320 (Figure S4 is not cited in the main text), but authors do not report lack of any clinical relevance in the main text or discuss this finding as limitation of their study/conclusion.

Reviewer #2 (Remarks to the Author):

Authors addressed most questions/concerns to improve the quality and impact of the manuscript. However, some issues are not adequately or insufficiently addressed and should be clarified prior publication:

1. Please harmonize presentation style for data in Fig. 1B and 1F or even better integrate them in one graph

We have adjusted the formatting for both graphs to look the same which is now included in updated figure 1. We have kept the graphs separate as they come from experiments done at separate times that went through separate analysis and normalization.

2. Despite the detailed response provided by the bioinformaticians, it remains a critical issue that the NOTCH signaling pathway is not ranked within the top 10 GO terms (Fig. 2B) and did not reach statistical significance. Authors should mention these facts in the main text and discuss them as potential limitations.

This is now mentioned in the manuscript (lines 178-182).

3. Revised Fig. 2F is confusing and detailed information is missing. What is the difference between the upper and lower graphs or Western blots? Western blots are incomplete: vimentin is missing for the upper and eIF4e for the lower. Moreover, protein expression patterns appear inconsistent with transcript levels. Though not explained at the figure legend * most likely indicates statistical significance ($p < 0.05$), which is questionable for 60 (lower left, vimentin) and 74 (lower right, E-cadherin), considering presented mean values + SEM. How many independent biological replicates were used to compute mean values + SEM?

There was an error in graph labeling that has been corrected. The data represents 3 replicates +SEM and follows the same trend as the data from qPCR analysis.

Figure S6A-B shows Vimentin and E-cadherin transcript levels (most likely label of the y-axis last graph in B is wrong) assessed by RQ-PCR. Presented data do not support in vitro findings and are biased by potential differences in the relative composition of the tumor by cancer and stromal cells. Hence, RQ-PCR cannot replace IHC or co-IF staining with tumor sections, and the conclusion on EMT and its reversal by NOTCH3 and its ligands in primary tumor and metastasis-derived cell lines are not supported by sufficient experimental evidence.

We agree that qPCR analysis presented in Figure S6 does not reveal differences in gene expression for EMT markers in subcutaneous tumors. We do not think it is possible to observe differences in EMT using this model; the experiment was done to demonstrate differential sensitivity of tumors to Notch3 but can not be used to study the mechanism of such sensitivity due to the limitations outlined by the reviewer. The suggestion of EMT playing a role in such mechanism is supported by in vitro data presented in figures 2F, 4C, D and 6D. This comment is now addressed in the body of manuscript (lines 323-330).

4. Please modify the statement at page 225-229 as DLL3 and PSEN1 did not reach $FDR < 0.05$ and other candidates (NOTCH3, HEYL, LCK) were not within the top 30 differentially essential genes concerning the relative dropout and not within the top 200 concerning FDR.

We have modified this statement accordingly and it is now included in the body of the manuscript (lines 229-232).

5. Authors stress several times in the rebuttal that metastasis-derived cells die upon NOTCH3 (or JAG2) silencing, which hamper further analysis (e.g. EMT-related morphology/gene expression or expression of other NOTCH receptors upon NOTCH3 or JAG2 silencing): “Cells derived from metastatic sites have dependency on Notch3 and Jag2 thus is it impossible to observe their phenotype/gene expression upon silencing – most the cells are dead”, “As discussed in point 3, it is not possible to obtain good quality expression data from the metastatic cells upon silencing of Notch3 as they are highly dependent on expression of this gene” and “we could not do much with metastatic cells when Notch3 is silenced as they are dying very quickly”. However, data presented in Fig. 3E and 3F demonstrate reduced growth rates over time, but not such a severe phenotype, and obviously good quality expression data were obtained to demonstrate silencing efficacy in Figure S1. Please clarify/explain conflicting statements.

The growth data is obtained by imaging using Incucyte Zoom Instrument and is capturing cells that are attached to the plate. Expression analysis is performed 48 hours post knockdown and when the cells are removed from the plates, it becomes obvious that the culture represents a mixture of live, dying, and dead cells. While Notch3 knockdown is very efficient and thus is easy to observe even under such challenging conditions, it is difficult to make a reliable conclusion about expression of other affected genes.

6. Figure S4 and Table S9 do not support any clinical relevance of downstream NOTCH components in the analyzed cohort, and Table S8 is inadequate to address my question. The crosstab should present association between relevant variables (e.g., pathological grading or lymph node metastasis) with high or low protein expression of JAG1, JAG2, HES1 or HEY1 to support experimental findings from cell culture and mouse models in patient tumor specimens. Table S9 is cited at line 320 (Figure S4 is not cited in the main text), but authors do not report lack of any clinical relevance in the main text or discuss this finding as limitation of their study/conclusion.

The expression of Notch pathway components in patients do not predict survival yet is indicative of status of the cells. In line with the in vitro data, cell lines derived from primary tumors express Jag1/Hes1 and low Jag2/Hey2, whether the patient they were derived from had a good or a bad prognosis. The pattern of expression changes in cells once they undergo metastatic transformation and can be visualized in tumors of higher grades, as shown in figure 5. Together with the differential dependency data, this finding suggests that novel therapies targeting Notch3/Jag2/Hey1 axis should be directed to patients that have metastatic tumors, rather than to preventing metastasis in patients with early grade tumors. We added this discussion to the manuscript.

The crosstab table has been modified according to reviewer suggestion.

REVIEWERS' COMMENTS:

Reviewer #2 (Remarks to the Author):

Authors addressed all questions in the rebuttal and adequately revised the manuscript.